# SLR1 inhibits MOC1 degradation to coordinate tiller number and plant height in rice

Zhigang Liao [1,2,4], Hong Yu [1,4], Jingbo Duan[1], Kun Yuan[1,2], Chaoji Yu[1,2], Xiangbing Meng[1], Liquan Kou[1], Mingjiang Chen[1], Yanhui Jing[1], Guifu Liu[1], Steven M. Smith[1,3] & Jiayang Li [1,2]

The breeding of cereals with altered gibberellin (GA) signaling propelled the 'Green Revolution' by generating semidwarf plants with increased tiller number. The mechanism by which GAs promote shoot height has been studied extensively, but it is not known what causes the inverse relationship between plant height and tiller number. Here we show that rice tiller number regulator MONOCULM 1 (MOC1) is protected from degradation by binding to the DELLA protein SLENDER RICE 1 (SLR1). GAs trigger the degradation of SLR1, leading to stem elongation and also to the degradation of MOC1, and hence a decrease in tiller number. This discovery provides a molecular explanation for the coordinated control of plant height and tiller number in rice by GAs, SLR1 and MOC1.

[1] State Key Laboratory of Plant Genomics and National Center for Plant Gene Research (Beijing), Institute of Genetics and Developmental Biology, The Innovative Academy of Seed Design, Chinese Academy of Sciences, Beijing 100101, China. [2] University of Chinese Academy of Sciences, Beijing 100039, China. [3] School of Natural Sciences, University of Tasmania, Hobart TAS 7001, Australia. [4] These authors contributed equally: Zhigang Liao, Hong Yu. Correspondence and requests for materials should be addressed to J.L. (email: jyli@genetics.ac.cn)

Tiller number and plant height are two important factors determining cereal plant architecture and grain yield[1]. Tiller number per plant determines panicle number, which is a key component of grain yield. The semidwarf genes, resulting in a shortened culm with improved lodging resistance and a greater harvest index, contributed to the "Green Revolution" in wheat and rice[2,3]. Tiller number and plant height are almost always inversely related in rice. Not only is dwarfism often associated with increased branching, but tillering also affects plant height in many mutants or transgenic plants[4–13]. Although progresses have been made in identifying separate mechanisms that control the development of tillers and plant height, no mechanism for their coordination has been revealed so far yet.

Numerous studies have shown that tiller number and development are regulated by a complex network of genetic, hormonal, and environmental factors. Among the genes involved in determining tiller number in rice, *MOC1* is one of the most important because it is required for the formation of axillary meristems (AM) and subsequent bud outgrowth[1,13]. The *MOC1* null mutant has no axillary buds and therefore no tillers, whereas *MOC1*-overexpressing plants are associated with an increased number of axillary buds and tillers[13]. Rice *MOC1* encodes a GRAS protein homologous to tomato Lateral suppressor (Ls)[14], and *Arabidopsis* LATERAL SUPPRESSOR (LAS)[15]. *MOC1* is expressed mainly in the axillary buds, and it has been found that TAD1 interacts with MOC1 and anaphase-promoting complex 10 (OsAPC10), targeting MOC1 for degradation in a cell cycle-dependent manner[6,16]. Recent studies have revealed that *MOC3/TILLERS ABSENT 1 (TAB1)*, an ortholog of *Arabidopsis thaliana* WUSCHEL (WUS), is also required to initiate the development of an axillary meristem (AM) in rice[17,18]. After AM formation, the *LAX PANICLE 1 (LAX1)* and *LAX2* genes are required for the maintenance of AM development[19,20]. Double-mutant analyses suggest that MOC1, LAX1, and LAX2 function in partially independent but overlapping pathways to regulate AM establishment and maintenance[19].

Phytohormones are another class of important factors determining bud fate. It has long been known that auxin inhibits the outgrowth of axillary buds, whereas cytokinin (CK) stimulates[21]. Strigolactone (SL) is a new class of plant hormone found to control branching[22,23]. Mutants deficient in SL biosynthesis or signaling all display more branching than the wild-type (WT). Tillering is also reported to be inhibited by gibberellin (GA) and promoted by brassinosteroid (BR)[11,24].

GAs regulate many developmental processes, such as seed germination, cell elongation, leaf expansion, flowering, and fruit development. DELLA proteins, characterized by a penta-peptide DELLA motif at the N terminus, are key components of GA signaling. Perception of GA by its receptor GIBBERELLIN INSENSITIVE DWARF 1 (GID1) leads to formation of a GID1-GA-DELLA complex, which triggers further DELLA interaction with an F box protein, SLEEPY 1 (SLY1) in *Arabidopsis* or GID2 in rice. The DELLA proteins are polyubiquitinated by the E3 ubiquitin-ligase SCF[SLY1/GID2] and then degraded by the 26 S proteasome[25–30]. Signaling of GA via GID1 and DELLA is well known to promote internode elongation, but it is not known if the corresponding reduction in shoot branching is a direct or indirect consequence of such GA signaling. In rice, *Arabidopsis*, tomato, wheat, and *Populus*, GA-deficient or GA-responsive mutants exhibit a shorter stature and higher branching than the wild types[11,31–36], but the mechanism underlying the phenotypes has not been discovered. In this study, we aimed to elucidate how GA signaling suppresses tiller number in rice by demonstrating that rice DELLA protein SLR1 interacts with MOC1 to inhibit the degradation of MOC1 and that GA promotes the degradation of SLR1 to bring about the inverse regulation of plant height and

tiller number. Thus, we establish that GA is directly involved in controlling tillering in rice.

## Results

**GA signaling inversely regulates tiller number and height**. While an inverse correlation between plant height and tiller number has been observed in GA mutants, it has not been systematically investigated. To explore this relationship further, we analyzed rice mutants with altered GA metabolism or signaling (Supplementary Fig. 1 and Supplementary Table 1) through measuring tiller number and plant height. We found that GA-deficient or GA-signaling mutants, including *semidwarf 1* (*sd1*), a GA biosynthesis mutant defective in *GA20ox-2, sdg* (a *gid1* weak allele), and *slr1-d1* (a *slr1* dominant allele), a GA-signaling mutant, produced more tillers and had a shorter stature than the WT plants (Fig. 1a–c; Supplementary Fig. 2a–f). Conversely, the *slr1* recessive mutant and a line overexpressing the GA biosynthesis gene *GA20ox-1* (*GA20-1OE*) exhibited fewer tillers and a taller stature (Fig. 1a–c; Supplementary Fig. 2g–i). To further investigate the role of GA, we produced transgenic plants with altered GA signaling. Consistent with the phenotypes of mutants, knockdown of *SLR1* by RNA interference (*SLR1*-RNAi) resulted in plants with fewer tillers and taller stature than the WT, whereas tiller number was increased by overexpressing *SLR1* fused with green fluorescent protein (*SLR1-GFPOE*) (Fig. 1d–f). To check whether GA can directly regulate tiller number, we treated 1-month-old seedlings for 1 month with GA or GA biosynthesis inhibitor paclobutrazol (PBZ). GA treatment decreased tiller number, whereas PBZ significantly increased tiller number and decreased plant height (Fig. 1g–i). These results establish that GA signaling consistently reduces the number of tillers produced while increasing plant height.

**GA regulates tiller bud outgrowth, not bud initiation**. The number of tillers produced could be determined either by the number of axillary buds or by the timing of their outgrowth. To address this question, we examined the axillary buds of GA-signaling mutants 30 days after germination (DAG). In the low-tillering *slr1* mutant, we could observe two buds in a shoot base and two in elongated upper internodes, which showed that the formation of tiller buds was normal in the *slr1* mutant (Fig. 2a; Supplementary Fig. 3a, b). Compared with WT plants, the length of the second axillary buds in the high-tillering *sd1* and *slr1-d1* mutants was longer, but shorter in the low-tillering *slr1* loss-of-function mutant (Fig. 2a; Supplementary Fig. 3c). These differences are consistent with the number of tillers observed at the heading stage. The GA-deficient *dwarf 18* (*d18*) mutant was then examined, because although this mutant is a severe dwarf, it does not produce more tillers than the WT at the heading stage. Interestingly, more axillary buds could be observed in the *d18* mutant than the WT at the seedling stage (Supplementary Fig. 3e). Apparently, the *d18* mutant does not go on to produce more tillers than the WT, because it suffers pleiotropic defects that limit subsequent tiller growth and development. To further confirm that GA signaling affects rice bud outgrowth, we examined *SLR1*-transgenic and GA-treated WT plants. In *SLR1-GFPOE* and PBZ-treated plants, the axillary buds emerged earlier and the length of the second buds was significantly longer than their controls, whereas it was opposite in *SLR1*-RNAi and GA-treated plants (Fig. 2c, e; Supplementary Fig. 4). Therefore, these observations are all consistent with GA suppressing the outgrowth of axillary buds but not affecting tiller buds initiation.

**Interaction between GA signaling and MOC1**. Since *MOC1* is a key determinant of axillary bud outgrowth and hence tiller

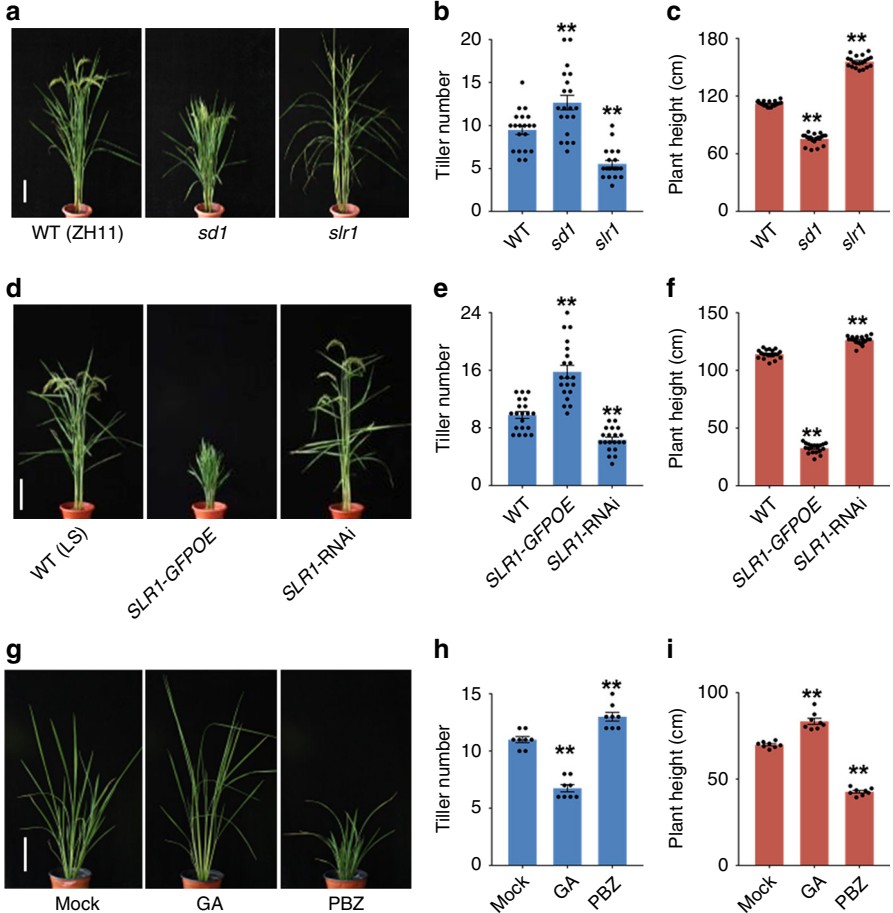

**Fig. 1** Tiller number and plant height are coordinated inversely by GA. **a** Appearance of the wild-type (WT, ZH11) and GA-signaling mutants at the heading stage. Scale bar, 20 cm. **b**, **c** Tiller number (**b**) and height (**c**) of plants shown in (**a**). **d** Appearance of *SLR1*-transgenic plants at the heading stage. Scale bar, 20 cm. **e**, **f** Tiller number (**e**) and height (**f**) of plants shown in (**d**). **g** Appearance of WT plants after GA and PBZ treatments for 1 month. Scale bar, 20 cm. **h**, **i** Tiller number (**h**) and height (**i**) of plants shown in (**g**). Asterisks indicate significant difference to the WT or mock (two-tailed Student's *t* test, **p < 0.01; mean ± s.e.m., n = 20 in (**b**, **c**, **e**, **f**), and 8 in (**h**) and (**i**). Source data are provided as a Source Data file

number, we asked if it is required for GA regulation of tillering. We showed that PBZ application did not stimulate tiller development in *moc1*, indicating that *MOC1* is indeed required for GA-mediated control of tiller growth and hence tiller number (Fig. 2g, h). Therefore, we further examined the levels of MOC1 and SLR1 proteins in extracts from the shoot bases of the WT and GA-signaling mutants. Compared with the WT, the levels of both MOC1 and SLR1 were increased in *sd1* but decreased in *slr1* (Fig. 2b), and both proteins were also increased in *slr1-d1* and *d18* mutants (Supplementary Fig. 3d, f). Similarly, MOC1 protein abundance was increased in *SLR1-GFPOE* transgenic plants but decreased in *SLR1*-RNAi plants (Fig. 2d). To further confirm these results, we examined the effects of GA and PBZ on MOC1 abundance. Application of GA promoted the degradation of SLR1, while the MOC1 protein level was also reduced (Fig. 2f). In contrast, PBZ treatment promoted the accumulation of both SLR1 and MOC1 (Fig. 2f). These results demonstrate that SLR1 and MOC1 protein abundance are coordinately regulated in the process of axillary bud outgrowth.

**MOC1 and SLR1 proteins physically interact.** DELLA proteins regulate various developmental processes by interacting with different proteins to affect their DNA binding, transactivation activities, or abundance[37,38]. Furthermore, both MOC1 and SLR1 are GRAS proteins, which in some cases are known to form homodimers or heterodimers[39,40]. The requirement for MOC1 in

regulating tiller number by GA raises the possibility that SLR1 might directly interact with MOC1. To test this hypothesis, we first examined their interaction in yeast two-hybrid assays and found direct interaction between MOC1 and SLR1 (Fig. 3a). In vitro pull-down assays also showed that purified glutathione S-transferase (GST) fusion protein GST-MOC1, but not GST alone, interacted with SLR1-GFP (Fig. 3b). To further confirm the interaction between MOC1 and SLR1 in vivo, we then performed bimolecular fluorescence complementation assays (BiFCs) by using the 35 S promoter-driven expression of the cyan fluorescent protein (CFP) N-terminal half (CN) fused to MOC1 (CN-MOC1). This was co-expressed with a fusion of the C-terminal half (CC) fused to SLR1 (CC-SLR1) in rice protoplasts. Strong CFP fluorescence signals were observed in the nucleus (Fig. 3c). However, no fluorescence signals were detected when CN-MOC1 was co-expressed with CC, or when CC-SLR1 was co-expressed with CN, indicating that MOC1 associates directly with SLR1 (Fig. 3c). Furthermore, we tested their interaction using a co-immunoprecipitation approach. For this purpose, we transiently co-expressed MOC1 fused with a FLAG tag (MOC1-FLAG) and SLR1 fused with GFP (SLR1-GFP) in rice protoplasts. MOC1-FLAG was co-immunoprecipitated when SLR1-GFP was pulled down from protoplast extracts with an anti-GFP antibody (Fig. 3d).

Next, we analyzed the MOC1 motifs necessary for interaction with SLR1. MOC1 comprises a short N-terminal region and a

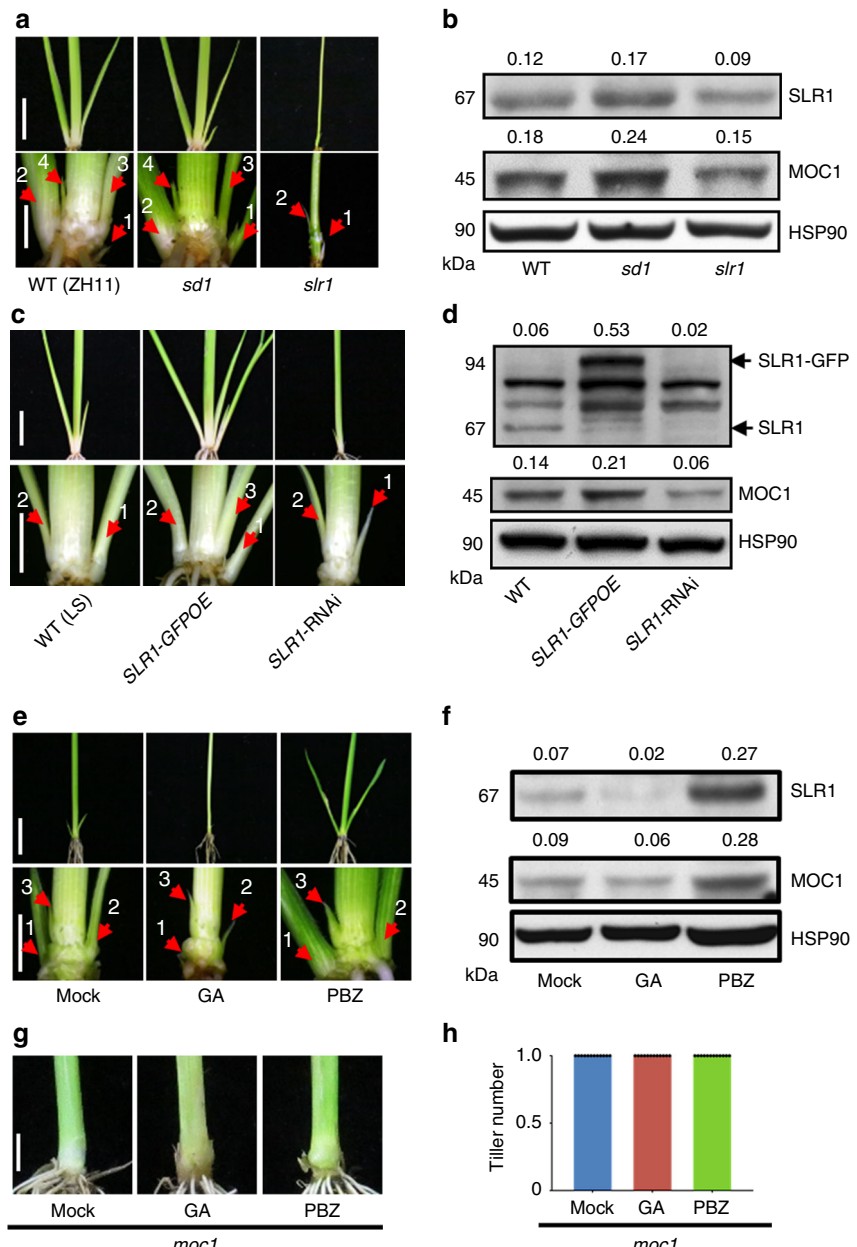

**Fig. 2** GA signaling inhibits tiller bud outgrowth and the protein level of MOC1. **a** Shoot bases of the WT (ZH11), *sd1*, and *slr1* seedlings at 30 DAG. The bottom row shows magnified images. The arrows indicate axillary buds or tillers. Scale bar, 2 cm (top) and 0.5 cm (bottom). **b** Protein levels of SLR1 and MOC1 in WT, *sd1*, and *slr1*, determined by protein blotting. Heat-shock protein 90 (HSP90), loading control. Values above panels indicate signal strength for SLR1 and MOC1 in arbitrary units determined by densitometry. **c** Shoot bases of WT (LS), *SLR1-GFPOE,* and *SLR1*-RNAi seedlings at 30 DAG. The bottom row shows the magnified images. Arrows indicate axillary buds or tillers. Scale bar, 2 cm (top) and 0.5 cm (bottom). **d** Protein blotting shows protein levels of SLR1 and MOC1 in WT (LS), *SLR1-GFPOE*, and *SLR1*-RNAi transgenic plants. HSP90, loading control. Values above panels indicate as in (**b**). **e** Effects of GA and PBZ treatments on tiller bud outgrowth. The bottom row shows the magnified images. Arrows indicate axillary buds or tillers. Scale bar, 2 cm (top) and 0.5 cm (bottom). **f** Effects of GA and PBZ treatments on SLR1 and MOC1 protein levels, determined by protein blotting. HSP90, loading control. Values above panels indicate as in (**b**). **g** Effects of GA and PBZ treatments on the tiller number of *moc1*. Scale bar, 0.5 cm. **h** Quantification of the tiller number shown in (**g**) (*n* = 12). The full scans of immunoblots are shown in Supplementary Fig. 12. Source data are provided as a Source Data file

large conserved GRAS domain of 380 residues, which can be subdivided into five distinct motifs: leucine-rich region I (LHRI), VHIID, LHRII, PFYRE, and SAW (Supplementary Fig. 5). Yeast two-hybrid assays were conducted using variously truncated MOC1 proteins. The results showed that deletion of the entire GRAS domain abolished the interaction of MOC1 with SLR1, but the truncations of any single motif were not sufficient to eliminate the interaction (Supplementary Fig. 5). In addition to the GRAS domain, SLR1 contains an N-terminal region, including the DELLA, TVHYNP, and polyS/T/V motifs (Supplementary Fig. 6a)[41]. Similarly, yeast two-hybrid assays revealed that deletion of the entire GRAS domain could abolish the interaction with MOC1, but the truncations of any single motif were not sufficient to abolish the interaction (Supplementary Fig. 6b, c). These results suggest that the physical interactions between MOC1 and SLR1 are complex but clearly occur both in vitro and in vivo.

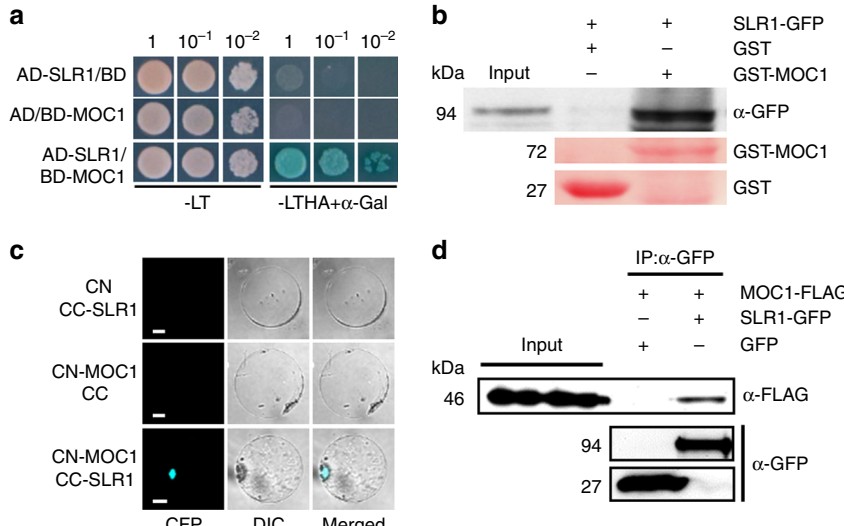

**Fig. 3** Interaction between MOC1 and SLR1. **a** Interaction between MOC1 and SLR1 in the Y2H assay. Transformed yeast cells were grown on synthetic media lacking Trp and Leu (-LT); Trp, Leu, His, and Ade (-LTHA); or α-Gal and 5-bromo-4-chloro-3-indoxyl-α-D-galactopyranoside, a chromogenic substrate for α-galactosidase. The -LTHA + α-Gal medium contained 40 μg·mL$^{-1}$ X-α-Gal. The numbers at the top indicate three serial dilutions. AD GAL4 activation domain, BD GAL4 DNA-binding domain. **b** Pull-down assay to test GST-MOC1 fusion protein or GST control interaction with SLR1-GFP using anti-GFP (α-GFP) immunoblotting. Red images show staining with Ponceau S. **c** BiFC analysis of the interaction between MOC1 and SLR1 in rice protoplasts. Scale bar, 20 μm. CC C-terminal portion of CFP, CN N-terminal portion of CFP. CFP fluorescence and differential interference contrast (DIC) are shown separately and merged. **d** In vivo interaction between MOC1-FLAG and SLR1-GFP, revealed by the co-immunoprecipitation assay. After immunoprecipitation with anti-GFP (α-GFP), precipitated proteins were probed with an anti-FLAG (α-FLAG) antibody. Lower panels show SLR1-GFP and GFP controls. Source data are provided as a Source Data file

**SLR1 inhibits the degradation of MOC1 independently of TAD1**. The observations that SLR1 and MOC1 bind to form a protein complex and that GA signaling leads to a decline in both protein levels raise the possibility that the amount of MOC1 could be regulated by SLR1. We thus tested whether SLR1 modulates the abundance of MOC1 by directly inhibiting its degradation. We therefore assayed the degradation of MOC1 in cell-free extracts prepared from the shoot bases of transgenic plants and mutants. For this purpose, the maltose-binding protein fused with MOC1 (MBP-MOC1) was expressed and purified from *Escherichia coli*, and then added to the extracts from WT and transgenic plants. We found that MOC1 proteins began to degrade in the extracts from WT and *GFPOE* plants within 30 min and its degradation was significantly reduced in the extracts from *SLR1-GFPOE* and *sd1* plants (Fig. 4a; Supplementary Fig. 7a). In contrast, MOC1 was more rapidly degraded in extracts from *GID1OE* plants, in which the amount of SLR1 was low (Fig. 4b). Moreover, addition of the His-Trx-SLR1 purified protein to the WT extracts could slow down the degradation of MBP-MOC1 (Fig. 4c; Supplementary Fig. 8), suggesting the direct inhibition of the MOC1 protein degradation by SLR1. The degradation of MOC1 in the extracts from *GID1OE* plants was slowed by the treatment of MG132 (Supplementary Fig. 7b), indicating that the degradation of MOC1 mediated by GA signaling is involved with the 26 S proteasome pathway. Taken together, these results demonstrated that SLR1 inhibits the degradation of MOC1.

Previous studies showed that MOC1 abundance is controlled by proteolytic degradation mediated by the APC/C$^{TAD1/TE}$ complex as its direct target, and the high tillering of the *tad1* mutant is caused by accumulation of MOC1[6,16]. The WT plants treated with PBZ were also dwarf and high tillering (Fig. 1g), similar to the phenotypes of the *tad1* mutant. This prompted us to test whether the SLR1-dependent inhibition of MOC1 degradation is abolished in the *tad1* mutant. We observed that the GA treatment could suppress tillering in *tad1*, but the PBZ

application could increase the tiller number (Fig. 4d–f). We then examined MOC1 protein levels in these plants and found that the MOC1 abundance in *tad1* plants was lower after GA treatment, but higher after PBZ treatment (Fig. 4g), which is consistent with the growth phenotype of *tad1* plants treated with GA or PBZ. These results suggest that SLR1 could inhibit the degradation of MOC1 without requiring the function of TAD1. Furthermore, we knocked down *SLR1* or overexpressed *GID1* in the *tad1* mutant, and observed a significant reduction in tiller number (Supplementary Fig. 9). Moreover, when we crossed *SLR1-GFPOE* with *TAD1OE*, the tiller number was similar to the WT plants (Supplementary Fig. 10). Taken together, these results support the model in which GA signaling regulates the level of SLR1 and then the MOC1 protein, which in turn regulates tiller number.

**GA controls plant height independently of MOC1**. Because SLR1 and MOC1 protein abundance are coordinately regulated and overexpressing *MOC1* results in plants with more tillers and shorter stature than the WT (Supplementary Fig. 11a–c), we therefore further asked whether SLR1 and MOC1 regulate plant height coordinately or separately. We examined the transcript and protein levels of SLR1 in the null mutant CRISPR/Cas9-created *MOC1* (*moc1$^{CR}$*) and WT seedlings. The results demonstrated that MOC1 had no obvious effect on the transcript and protein levels of SLR1 (Fig. 5a; Supplementary Fig. 11d). Cell-free degradation assays further showed that Hix-trx-SLR1 is rapidly degraded within 30 min (Fig. 4c; Supplementary Fig. 10e), which is similar to all the five *Arabidopsis* DELLA proteins[42], but MOC1 showed no effect on the degradation of SLR1 (Supplementary Fig. 11e). Furthermore, we showed that PBZ application was able to reduce the height of *moc1,* whereas GA treatment increased the height (Fig. 5b, c), suggesting that GA controls plant height independently of MOC1. To confirm this hypothesis, we created a *GID1* null mutant by CRISPR/Cas9 (*gid1$^{CR}$*) and crossed it with *moc1$^{CR}$*. The double mutants showed that the tiller

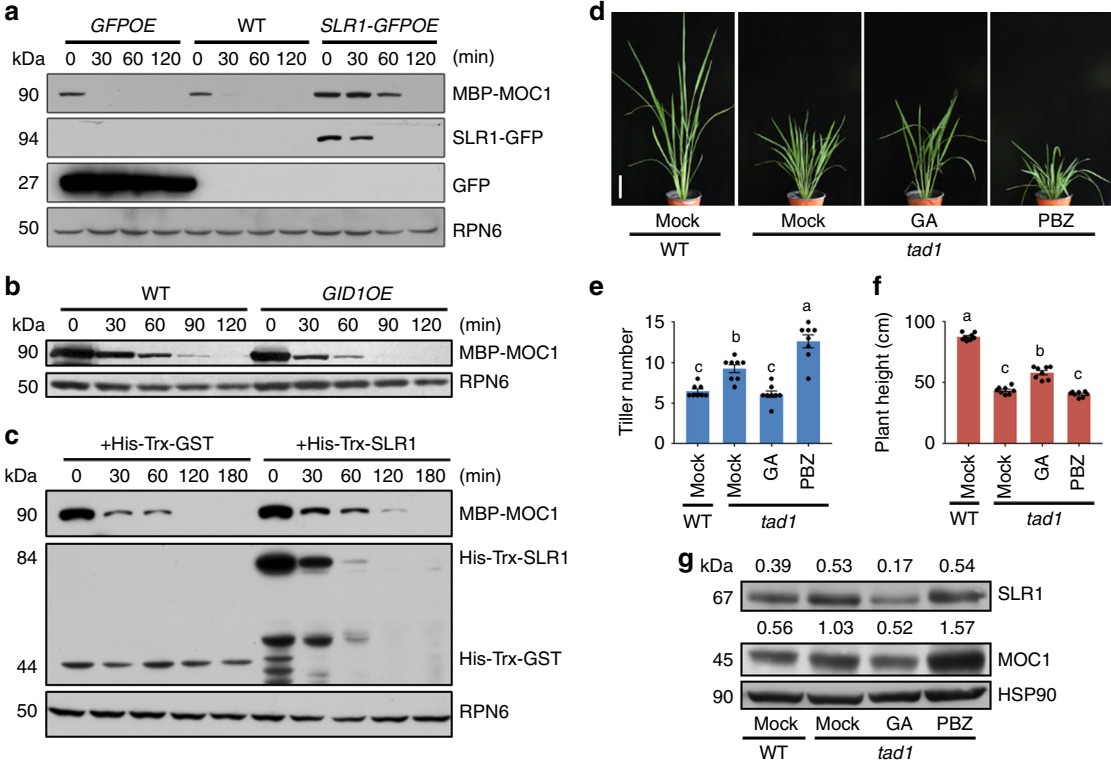

**Fig. 4** SLR1 inhibits the degradation of MOC1 independently of TAD1. **a** In vitro cell-free protein degradation assay, showing degradation of MBP-MOC1 in extracts from WT, *GFPOE* and *SLR1-GFPOE* transgenic plants. Immunoblots were probed with anti-MBP (α-MBP). Ribosomal protein 6 (RPN6), loading control. **b** In vitro cell-free protein degradation assay, showing degradation of MBP-MOC1 in extracts from WT and *GID1OE* transgenic plants. Immunoblots were probed as in (**a**). **c** In vitro cell-free protein degradation assay, showing that degradation of MBP-MOC1 was inhibited by His-Trx-SLR1, but not His-Trx-GST. Immunoblots were probed as in (**a**). **d** Effects of GA and PBZ treatments on *tad1*. Scale bar, 10 cm. **e**, **f** Quantification of the tiller number (**e**) and height (**f**) of plants shown in (**d**). Different lowercase letters indicate significant differences (Tukey's HSD test, $p < 0.01$; mean ± s.e.m., $n = 8$). **g** Immunoblot analysis of SLR1 and MOC1 proteins in shoot basal tissues (0–0.5 cm) after treatments. Tissue samples were analyzed by protein blotting using α-SLR1 and α-MOC1. HSP90, loading control. Values above panels indicate signal strength for SLR1 and MOC1 in arbitrary units determined by densitometry. The full scans of immunoblots are shown in Supplementary Fig. 12. Source data are provided as a Source Data file

number was identical to *moc1*$^{CR}$, while the plant height was similar to *gid1*$^{CR}$ (Fig. 5d–i). Taken together, these results suggest that the GA control of tiller number but not plant height depends on MOC1.

## Discussion

We propose a model to illustrate how GA signaling via SLR1 directly coordinates plant height and tiller number (Fig. 5j). We present new evidences that GA signaling controls tiller number by directly controlling the level of MOC1 protein, which is protected from degradation through the interaction with the DELLA protein SLR1. GAs trigger the degradation of SLR1 and lead to the degradation of MOC1, resulting in a decrease in tiller number. At the same time, GA signaling acting through SLR1 can promote shoot elongation independently of MOC1, as shown in the *gid1*$^{CR}$ *moc1*$^{CR}$ double mutant. The two processes are therefore inextricably linked through SLR1, but apparently act upon different downstream targets. Such a system therefore ensures coordination of plant height and tiller number by GA signaling and *MOC1*, while also providing plasticity that enables fine tuning by other factors. We therefore now have a clear molecular understanding of how GAs elegantly regulate rice shoot architecture and how this has enabled us to produce the high-yielding semidwarf rice varieties upon which we now depend.

Increased tiller number is usually associated with decreased height, and vice versa. However, the relationship is plastic, as the final height and tiller number depend on numerous endogenous and environmental factors, such as phytohormone signaling of auxin, CKs, SLs, and BRs, together with mineral and photoassimilate availability, and resource partitioning. Both GA and SL inhibit the outgrowth of lateral buds and stimulate stem elongation, so it will be important to understand how the actions of these hormones are coordinated at the level of biosynthesis and signaling[43,44]. It will also be important to investigate the integration of hormone signaling with resource allocation, since studies in wheat and sorghum have also shown that carbohydrate partitioning also plays a key role in determining whether lateral buds develop into tillers and the final architecture of the shoot[45,46].

Although MOC1 has been known for many years, its action mode still remains unknown. Expression of *MOC1* is initially restricted to a few epidermal or subepidermal cells in the leaf axils, then in the axillary meristems, and subsequently the entire tiller buds, including axillary leaf primordia and young leaves[13]. This appears to imply pleiotropic functions for MOC1 at different stages of development. It is also known that MOC1 homologs in other cereals can influence panicle or ear development[47]. Previous studies in rice showed that MOC1 is essential for the formation of AMs and buds[13]. In future, examination of the in vivo interaction between native MOC1 and SLR1 at different developmental stages could uncover more specific regulatory mechanisms. It was found that TAD1 interacts with MOC1 and OsAPC10, targeting MOC1 for degradation in a cell cycle-dependent manner[6]. In

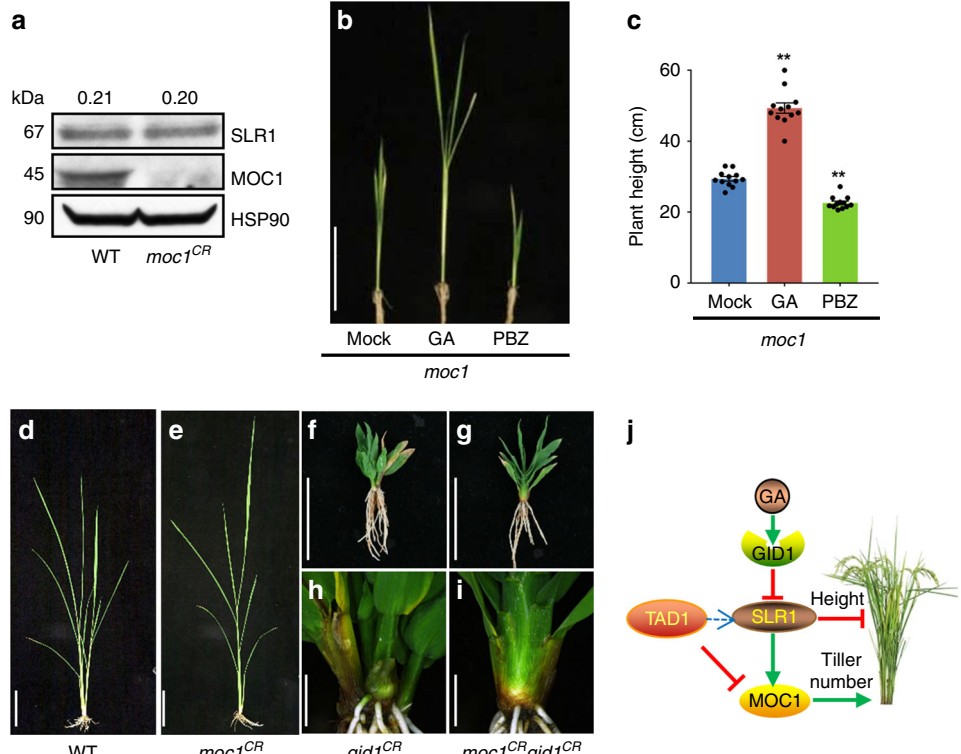

**Fig. 5** Controls of plant height by GA independently of MOC1. **a** Protein levels of SLR1 and MOC1 in the WT and *moc1*[CR], determined by protein blotting using α-SLR1 and α-MOC1. HSP90, loading control. Values above panels indicate signal strength for SLR1 in arbitrary units determined by densitometry. The full scans of immunoblots are shown in Supplementary Fig. 12. **b** Effects of GA and PBZ treatments on plant height of *moc1*. Scale bar, 20 cm. **c** Quantitation of the plant height shown in (**b**). Asterisks indicate significant difference to the mock (two-tailed Student's *t* test, **$p < 0.01$; mean ± s.e.m., $n = 12$). **d–i** Phenotypes of the WT, *moc1*[CR], *gid1*[CR], and *moc1*[CR] *gid1*[CR] after 2 months of growth. **h**, **i** show the magnified shoot bases of (**f**) and (**g**), respectively. **d**, **e** Scale bar, 20 cm; (**f**, **g**), scale bar, 5 cm; (**h**, **i**), scale bar, 0.5 cm. **j** A model of GA signaling and MOC1 regulation of plant height and tiller number in rice. Source data are provided as a Source Data file

contrast, the results presented here further show that the GA-dependent degradation of MOC1 occurs independently of TAD1 and apparently controls the outgrowth of buds (Fig. 2). We also observed that *MOC1*-overexpressing lines were infertile, whereas *tad1* mutants were fertile[6,16]. This raises the possibility that TAD1-independent regulation of panicle development by MOC1 occurs through interaction with SLR1 or other proteins. The identification of new proteins interacting with MOC1 may provide new insights to understand the panicle development in rice.

The *tad1* mutant has a higher level of SLR1 protein and a shorter plant height compared with the WT (Fig. 4f, g). We also noticed that *GID1OE* but not *SLR1*-RNAi could rescue the dwarf phenotype of *tad1* (Supplementary Fig. 9a, c). The plant height of *tad1* treated with GA is higher than the control, but still lower than the WT without GA treatment (Fig. 4d, f), whereas the protein level of SLR1 was lower in *tad1* after GA treatment compared with the WT (Fig. 4g). These results suggested that TAD1 is likely to crosstalk with the GA-signaling pathway and SLR1 may not be the only regulator of plant height (Fig. 5j). Moreover, the increases of MOC1 protein levels are also stronger in *d18* than in *slr1-d1*, which is consistent with the observation that *d18* exhibits longer axillary buds than *slr1-d1*. The increase or decrease of SLR1 and MOC1 protein levels in WT treated with PBZ or GA are stronger than the increase or decrease in *sd1* or *slr1* mutants. It is possible that other factors in the GA pathway may affect SLR1 and MOC1.

Our discoveries demonstrate that it is important to determine how the SLR1–MOC1 mechanism is influenced by other genes known to regulate tiller number and plant height. Importantly,

these results will provide opportunities to optimize genetic approaches to achieve improved outcomes in breeding optimal tiller number and plant height in rice by selecting particular combinations of alleles for TAD1, MOC1, and/or GA-signaling components.

## Methods

**Plant materials and growth conditions**. The rice plants (*Oryza sativa* L. spp. *Japonica*) used in this study, including the wild-type plants, the GA-related mutants (Supplementary Table 1), and the relevant transgenic plants, were grown in the experimental fields at the Institute of Genetics and Developmental Biology in Beijing or in Sanya, Hainan Province during the natural growing seasons. In all experiments, seeds were sown in soil mixture in plastic pots. At the 3–8 leaf stage, seedlings were prepared for observation or experimental treatment. In chemical treatment experiments, seedlings were supplied through the roots with $10^{-5}$ M GA (Sigma-Aldrich, Catalog no. G1025) or PBZ (Phyto Technology Laboratories, Catalog no. P687) for 3 weeks or 1 month.

**Generation of transgenic rice plants**. For the generation of rice plants over-expressing proteins, the full-length cDNA coding sequences (CDSs) of *SLR1* or *GID1* were amplified by PCR using specific primers (Supplementary Table 2), and then *SLR1* was inserted into the AHLG vector via *Sma*I and *Xba*I digestion and *GID1* into the pTCK303[48] vector via *Bam*HI and *Spe*I digestion. For generation of the *SLR1*-RNAi plants, two inverted fragments of *SLR1* were amplified by PCR, digested with *Bam*HI and *Kpn*I and with *Spe*I and *Sac*I, and cloned into the pTK303 vector. For generation of CRISPR/Cas9 transgenic plants, the guide sequences were inserted into the gRNA expression vector backbone prU6-gRNA using annealed oligonucleotides following the instructions of a Cas9/gRNA Construction Kit (ViewSolid Biotech, Beijing, Catalog no. VK005-01). The resulting constructs were transfected into *Agrobacterium tumefaciens* strain EHA105 for transformation of the wild type or *tad1*, respectively.

**Antibody preparation and immunoblots**. A DNA fragment of *SLR1* (encoding amino acid residues 1–278) or full-length cDNA of *MOC1* were cloned into the pET-28a vector (Novagen, Catalog no. 69864-3). The recombinant proteins were expressed in *E. coli* BL21 (DE3) cells and purified by Ni-sepharose (GE Healthcare, Catalog no. GE17-5268-01). The purified proteins were used to raise polyclonal antibodies in rabbit. The resulted antibodies were purified through an IgG-affinity chromatography column (GE Healthcare, Catalog no. 17508001) before use. To immunodetect endogenous SLR1 and MOC1 levels in plants, shoot bases (0–0.5 cm aboveground) were collected and total proteins were extracted in extraction buffer (50 mM sodium phosphate, pH 7.0, 150 mM NaCl, 10% (v/v) glycerol, 0.1% (v/v) Nonidet P-40, and 1× complete protease inhibitor cocktail) for protein blotting. Primary antibodies anti-SLR1 and anti-MOC1 were used at a 1:1000 (v/v) dilution, and anti-HSP90 (LifeSpan BioSciences, Catalog no. LS-C178777) at a 1:20,000 (v/v) dilution. The specificity of anti-SLR1 polyclonal antibodies was validated using SLR1-GFP proteins in the rice protoplast extracts (Supplementary Fig. 13a). The specificity of anti-MOC1 polyclonal antibodies was validated using proteins extracted from *moc1* and *moc1CR* mutant plants (Supplementary Fig. 13b, c).

**Protein interactions**. For yeast two-hybrid assays, full-length *SLR1* and the truncations of *SLR1* were amplified using specific primers (Supplementary Table 2) and fused with GAL4 BD in the pGBKT7 vector (Clontech, Catalog no. 630443). Full length and truncations of *MOC1* were amplified using specific primers (Supplementary Table 2) and fused with GAL4 AD in the pGADT7 vector (Clontech, Catalog no. 630442). Interactions in yeast were tested on the SD/-Trp/-Leu/-His/-Ade/X-α-Gal medium. For pull-down assays, the recombinant GST-MOC1 and GST proteins were expressed in *E. coli* BL21 (DE3) cells and purified through a GSTrap HP column (GE Healthcare, Catalog no. 17-5282). Then 1 μg of recombinant GST-MOC1 or GST proteins bound to glutathione beads (GE Healthcare, Catalog no. 17-5279) were incubated with 1 mL of total proteins extracted from *SLR1-GFPOE* calli by extraction buffer (see above) at 4 °C for 2 h with gentle rotation, and washed three times with extraction buffer. Their interaction was detected by a monoclonal anti-GFP antibody (Roche, Catalog no. 11814460001) at a 1:5000 (v/v) dilution. For the BiFC analysis, the cDNA of *SLR1* and *MOC1* were amplified with primers (Supplementary Table 2) and cloned into pSCYCE and pSCYNE vectors[49] containing either C- or N-terminal portions of the enhanced cyan fluorescent protein. The resulting constructs were used to transform using protoplasts by polyethylene glycol (PEG)-mediated transformation method[50]. After incubation in the dark for 14 h, the CFP fluorescence was observed with a confocal laser-scanning microscope (FluoView 1000, Olympus). For co-immunoprecipitation assays, full-length *SLR1* and *MOC1* were amplified (Supplementary Table 2) and inserted into SE and SF vectors[6], respectively. The resulting transient expression constructs were used to transform protoplasts as described above. After incubation for 14 h, proteins were extracted with the phosphate extraction buffer (see above) and used for co-immunoprecipitation experiments as described previously[6]. Their interaction was detected by a monoclonal anti-FLAG antibody (Abmart, Catalog no. M2008) at a 1:10,000 (v/v) dilution.

**Cell-free protein degradation assay**. For cell-free protein degradation assays, full-length *MOC1* was amplified using primers (Supplementary Table 2) and cloned into the pMAL-c2x vector (NEW ENGLAND BioLabs, Catalog no. E8000S). Recombinant MBP-MOC1 proteins were expressed in *E. coli* BL21 (DE3) cells and purified through Amylose Resin (NEW ENGLAND BioLabs, Catalog no. E8021S). Seedlings of *GFPOE*, *SLR1-GFPOE*, *sd1*, *GID1OE*, and the corresponding WT were grown in a SANYO MLR Plant Growth Chamber (MLR-351) for 14 days at 28 °C during daytime and 25 °C at night with a 16:8 (day:night) photoperiod. The shoot bases (0–0.5 cm aboveground) were collected, and total proteins were extracted with degradation buffer (25 mM Tris-HCl, pH 7.5, 10 mM NaCl, 10 mM MgCl$_2$, 4 mM PMSF, 5 mM dithiothreitol, and 10 mM ATP)[42] and adjusted to equal concentrations with the degradation buffer. Then 100 ng of purified MBP-MOC1 or His-trx-SLR1 protein was incubated in 100-μL extracts (containing 500 μg of total proteins) for the individual assays. The extracts were incubated at 28 °C, and samples were taken at the indicated intervals for protein blotting. Anti-MBP (NEW ENGLAND BioLabs, Catalog no. E8032), anti-His (Abmart, Catalog no. M20020), and anti-RPN6 (Enzo Life Sciences, Catalog no. BML-PW8370) antibodies were used at a 1:20,000 dilution.

**RNA isolation and qPCR**. Total RNAs from the organs examined were extracted using a TRIzol kit according to the user's manual (Invitrogen, Catalog no. 12183555). Total RNAs (2.5 μg) were treated with DNaseI and used for cDNA synthesis with a SuperscriptIII RT kit (Invitrogen, Catalog no. 18080093). qPCR experiments were performed with gene-specific primers (Supplementary Table 2) in the reaction system of SsoFast™ EvaGreen supermix (Bio-Rad, Catalog no. 172-5201AP) on the CFX96 Real-time system (Bio-Rad, Catalog no. 184-5384) following the manufacturer's instructions. The rice *OsUbiquitin* gene (*LOC_Os03g13170*) was used as an internal control.

**Reporting summary**. Further information on research design is available in the Nature Research Reporting Summary linked to this article.

## Data availability

The authors declare that all data supporting the findings of this study are available within the paper or its supplementary files or are available from the corresponding authors upon request. The source data underlying Figs. 1b, c, e, f, h, i, 2h, 3b, d, 4a, c, e, f, and 5c, and Supplementary Figs. 2b, c, e, f, h, I, 4a–b, 7a–b, 8, 9b–c, 10b–c, and 11b–e are provided as a Source Data file.

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

## Acknowledgements

We thank Prof. Qian Qian (China National Rice Research Institute, Chinese Academy of Agricultural Sciences) for providing the mutants, *sdg*, *slr1-d1*, and *slr1*, and Prof. Chengcai Chu (Institute of Genetics and Developmental Biology, Chinese Academy of Sciences) for providing the mutants, *d18*, *sd1*, and *GA20-1OE*. This work was supported by grants from the National Natural Science Foundation of China (No. 31788103, No. 91635301), the Ministry of Science and Technology of China (No. 2016YFD0101800), the Beijing Short-Term Recruitment Program of Foreign Experts, and the Chinese Academy of Sciences President's International Fellowship Initiative (2018VBA0025).

## Author contributions

Z.L. and H.Y. designed research, analyzed the data, and wrote the paper; Z.L. performed experiments; J.D., K.Y., C.Y., X.M., L.K., M.C., Y.J., and G.L. performed some of the experiments. S.M.S. and J.L. analyzed the data and wrote the paper. J.L. supervised the project, designed research, analyzed the data, and wrote the paper.

## Additional information

**Competing interests:** The authors declare no competing interests.

