## [Peer Review File · Nature Communications]

Reviewers' comments:

Reviewer #1 (Remarks to the Author):

As mentioned in the introduction part of this paper, tiller number and plant height are two important factors determining cereal plant architecture and grain yield, as mentioned in the introduction of this paper. Thus, it should be important to study the molecular relationship between these traits, although there is no remarkable study so far. However, no study has been done on the relationship between these two traits at molecular level. In this paper, Liao et al. focused on two genes: SLR1, and MOC1, that is, SLR1 that functions to suppress gibberellin (GA) signaling resulting in dwarfism, and MOC1 that is a tiller regulating component. By morphological, biochemical, and molecular biological analyses, the authors revealed that SLR1 interacts with MOC1 and inhibits the degradation of MOC1, whereas GA causes stimulates the SLR1 degradation to thus bring about results in the inverse regulation of plant height and tiller number.

As mentioned, it is an indubitable fact that the integrated study on the molecular mechanism(s) controlling the tiller number and plant height should be important from the view point not only of in basic science but also of in practical rice breeding. In this context, the studies by their study on revealing such molecular mechanism could be a epoch making pioneering work to directly reveal such molecular mechanism. However, this paper has many concerns, solicitudes, and problematic issues for publication in a high impact journal. I listed my comments those as follows.

1. In Fig. 2a, I could not figure out that the length of axillary buds of sd1 plant is longer than that of the control plant. Similarly, I can not exactly figure out the exact phenomenon phenotypes they mentioned about Fig. 2a (WT vs. slr1) or 2c (WT vs. SLR1-RNAi). In Sup. Fig. 3a, I realized that the right axillary bud at the right of the slr1-d1 plant was longer than that of WT, but not in the case of the left side. Comparison Characterization of the length of axillary buds is a very important part for this paper study, I would like to suggest the authors to present the state of axillary buds as like in Sup. Fig. 3c, by which readers could easily figure out the difference in the length of axillary buds.

2. In Fig. 2d, the authors did not mention about the SLR1 protein level, in Fig. 2d even though. In Fig. 2d, four bands with different sizes were recognized by the SLR1 antibody. Among them, the authors suggested that the biggest and smallest bands correspond to SLR1-GFP and SLR1, but no without explanation for the other middle two bands in the middle. Are these non-specific bands? Why were these intermediate bands were not detected in the SLR1 blot experiments for other plants in Fig. 2b or f?. I want to would like to see the full-scans results of SLR1 blotting in Fig. 2b, 2f, Sup. Fig. 3b, and Sup. Fig. 3f, as like Fig. 2d.

Although figures in Sup. Fig. 9-14 are very good for correct discussion, After writing the above comment, I found they the "full-scan" (the authors used the word, full-scan, but the photos presented in Sup. Fig. 9-14 are still closed view around the SLR1 bands in Sup. Fig. 9-14.

Although these figures are very good for correct discussion, they have problems, and. Actually, these are not real full-scan photos, but only closed up the neighborhood of SLR1 band, whereas the authors did not mention about these supplemental figures in the text at all. Furthermore, I could not recognize border lines between SLR1 and MOC1 blots in Sup. Fig. 9, 12, and 13, which makes me difficult to separate two photos blotted by the SLR1 or MOC antibody.

Anyway, my above question is not settled by this full-scan result of Sup. Fig. 9b. On the contrary, after seeing these full-scan results, I had some other questions about the immunoactive bands. Actually, there are other immunoactive bands, which were not found in Fig. 2. The author just focused on the bands in the red box in Sup. Fig. 9 to present in Fig. 2. However, how did the authors recognize which band corresponds to the real SLR1/MOC1? In Sup. Fig. 9a, for example, there is one strong band showing higher mobility than the SLR1 band squared by a red box (I am not sure whether the band with his higher mobility band really blotted by the SLR1 antibody), of the intensity of which the intensity is not different among the three samples. Then, how did the authors conclude that SLR1 really correspond to the bands squared by red line in the red box, but not to the lower bands? There are I have similar concerns for similar situations in

other results not only in the case of SLR1 blot but in the results of MOC1. The authors should show a clear give a reason why the bands squared by a red line in the red box really correspond to the exact SLR1/MOC1 protein.

3. Among Fig. 2 and Sup. Fig. 3, the most important result could be Fig. 2e and f (Mock vs. GA- or PBZ-treatment) for their argument, I guess. Again, I could not find a clear difference in the state of axillary buds or the MOC1 protein level among these plants (although I recognized delicate differences between Mock vs. GA- or PBZ-treatment). I would like to see a clearer difference in the axillary bud length and clearer difference in protein blotting results of MOC1 and SLR1, as because these are the fundamental observations for their argument of in this paper.

4. About Sup. Fig. 4a, the authors mentioned that the numbers indicate the position of amino acids. I confirmed the correspondence between the numbers in Sup. Fig. 4a and the amino acid residue of rice MOC1. The 61st amino acid of MOC1, which contains 441 amino acids, is Leu whereas, although this figure says N61. The 75th amino acid is Ile, while this figure says C75. Similarly, I found no correspondence between the amino acid position of SLR1 and the number problem is found in Sup. Fig. 5b: t. For example, the 230th amino acid of SLR1 is C, while the figure says N230, and the 169th is V but the figure says C169.

5. The authors mentioned that LHRI and VHIID motifs are important for interaction with MOC1. However the result in Sup. Fig. 4b shows that MOC1 Δ LHRI formed the strongest colony among the deletion series of SLR1, whereas MOC1 Δ VHIID formed the faintest one. Furthermore, although the authors mentioned that the LHRI motif of SLR1 is important for interaction with MOC1, I cannot find any difference in the colony formation between SLR1 Δ LHRI and other deletion series of SLR1 (Sup. Fig. 5c). The clearest effect was seen in SLR1N230 (located in polyS/T/V, but its position may be incorrect) and SLR1C75 (intervening region between DELLA and TVH) in Sup. Fig. 5b. I would like to strongly recommend that the authors redo a yeast two hybrid experiment for more quantitative way a better quantification by a liquid assay, not by colony assay.

6. The results presented in Fig. 4a and b are very important observations for the conclusion of this paper this work. As the authors have some other suitable plants for this experiment, such as *sd1*, *slr1*, *slr1-d1*, and GA20-1OE, I would like to see the results of the same experiment using these plants to confirm their conclusion. I wonder why they used plant another plant, GID1OE, for this experiment, but not plants such as *sd1*, *slr1*, *slr1-d1*, and GA20-1OE and not the above plants, which were already analyzed before to show a relationship between GA signaling and axially bud elongation. I would like to see the results for these plants in the same experiment.

7. Based on the results in Fig. 4c-f and Sup. Fig. 7, the authors concluded that SLR1 inhibits the degradation of MOC1 independently of TAD1. I cannot agree with this conclusion only by these results, because the amount of SLR1 is higher in *tad1* than in WT (Fig. 4f), demonstrating that the *tad1* mutation causes higher accumulation of SLR1, and resulting in higher accumulation of MOC1. The authors should clarify the molecular/genetical relationship between TAD1 and SLR1 by other approaches.

Reviewer #2 (Remarks to the Author):

This is a well-written manuscript that presents novel data to explain the observation of increased tillering (branching) in gibberellin (GA)-deficient plants. Such plants accumulate DELLA protein and here it is shown that the rice DELLA, SLR1, interacts with and stabilises the tillering promoter MOC1. The interaction between SLR1 and MOC1 is demonstrated convincingly using different methods. It is also shown that MOC1 protein level is positively correlated with SLR1 content using numerous lines with altered SLR1 content and chemical treatments. Importantly, by crossing lines in which MOC1 and the GA-receptor GID1 are knocked down by CRISPR/Cas9, they show that GA-reduction of tiller number (but not its promotion of plant height) is dependent on MOC1 function. These are important results of general interest. However, I have a question about the overall interpretation. The authors suggest that SLR1 promotes MOC1 function by stabilising it. However, the interaction of SLR1 with MOC1 would be expected to affect its function. Another interpretation is that SLR1 and MOC1 act as a complex to promote branching. There are many examples of

DELTA proteins acting as transcriptional co-activators in combination with transcription factors. I don't believe that the results presented in this manuscript discount this possibility. One aspect of the work I did not find convincing was the yeast 2-hybrid assays to determine the protein motifs in MOC1 and SLR1 that are necessary for the interaction. Clearly a lot of work was put into this, but the results seem unclear. For example, the authors suggested that the LHR1 motif in MOC1 is important for the interaction with SLR1, yet in Fig S4, the Δ LHR1 sequence appears to produce one of the strongest interactions. I think the statement on page 9 that LHR1 and VHIID motifs of MOC1 are essential for the interaction should be reconsidered. In Figs S4 and S5, it is unclear what the C-terminal fragments refer to. Does the number in for example MOC1C75 refer to the numbers of residues missing from the C terminus or to a fragment containing amino acids from 75 to the C terminus? It would be helpful to the reader, particularly to one not familiar with the GA-signalling field, if the mutants used in this study are explained. For example, it should be explained that the *slr1* recessive mutant results in loss of function due to mutation in the GRAS domain while the *slr1-d1* dominant mutant causes stabilisation and gain of function due to an N-terminal mutation. It could also be pointed out that the *sd1* mutant is GA-deficient due to knock-down or loss of GA2ox2. Fig S1 could be helpful for this purpose.

Reviewer #3 (Remarks to the Author):

In rice, tiller number and plant height are usually inversely regulated. Increase in GA levels promotes plant height and reduces tiller number, whereas impaired GA signaling or biosynthesis reduces plant height and increases tiller number. SLR1 is known to suppress plant height and MOC1 to promote tiller number. Authors of this manuscript explored the mechanism of tiller number suppression and plant height promotion by GA. They found that SLR1 interacts with MOC1 and inhibits the degradation of MOC1. With increased levels of GA, levels of both SLR1 and MOC1 were shown to be slightly reduced and proposed as a mechanism for increase in plant height and decrease in tiller number. Various rice mutants and GA and PBZ (GA biosynthesis inhibitor) treatments were employed in all experiments. The pathway of GA positively regulating *GID1* (GA receptor), which negatively regulating SLR1, which positively regulating MOC1 seems to be supported by some data. However, the role of TAD1 in the regulation of MOC1 and SLR1 is not completely supported by data. Specific comments are described as follows:

1) Fig. 2: The increase in both SLR1 and MOC1 levels in *sd1* (the GA receptor) mutant and decrease in both SLR1 and MOC1 levels in *slr1* mutant (Fig. 2b) do not seem to be as significantly as treatment of WT with PBZ and GA (Fig. 2f), suggesting that degradation of the SLR1-MOC1 complex by GA may involve other factors.

2) Fig. 4: The E coli-expressed MBP-MOC1 fusion protein was incubated with cell extracts from SLR1-GFP- and *GID1*-overexpression transgenic plants (Fig. 4a,b). MBP-MOC1 levels were shown to be stabilized in SLR1-GFP-overexpressing extracts but degraded rapidly in *GID1*-overexpressing extracts. However, the levels of overexpressed SLR1-GFP and *GID1* in cell extracts throughout the incubation time were not provided, making it uncertain whether the stabilization or degradation of MBP-MOC1 was indeed caused by overexpression of SLR1-GFP and *GID1*. Cell extract from GFP-overexpressing transgenic plants should also be provided as a control. It is possible that non-specific protein degradation by general proteases may occur, and overexpression of certain proteins may dilute out these proteases present in cell extracts.

Since GA promotes the degradation of SLR1 via the 26S proteasome pathway, *GID1*-overexpression should increase the degradation of SLR1. Whether the 26S proteasome inhibitor MG132 can stabilize the MBP-MOC1 complex should be tested to demonstrate the specificity of MOC1 degradation by the *GID1*-mediated 26S proteasome pathway.

The pathway proposed in Fig. 5j shows that TAD1 interacts and degrades MOC1, which acts independently of the GA signaling pathway. However, in the *tad1* mutant, SLR1 level was increased (Fig. 4f) and plant height was reduced as compared with the WT (Fig. 4e), indicating TAD1 suppressed SLR1 level. Additionally, treatment of *tad1* mutant with GA reduced SLR1 level by 2-fold (Fig. 4f) but still reduced plant height (Fig. 4e). These results indicate that TAD1 may crosstalk with the GA signaling pathway in regulating plant height, and SLR1 may not be the only negative regulator of plant height. The notion is supported by data in Supplementary Fig 6b,c, in which tiller number is increased and plant height is reduced in *tad1* mutant, but both not in *tad1/GID1-OE*. The notion is also supported by some data in Supplementary Fig. 7a, b.

3) Fig. 5: Since the interaction between SLR1 and MOC1 stabilizes the two proteins, why the accumulation of MOC1 is abolished but that of SLR1 remains the same in the *moc1/crispr* mutant as compared with the WT (Fig. 5a)?

4) In summary, some data are contradictory and inconsistent as mentioned above, and the proposed model in Fig. 5 j may need modification.

Point-by-point response to Reviewers

Response to the comments of Reviewer 1:

As mentioned in the introduction part of this paper, tiller number and plant height are two important factors determining cereal plant architecture and grain yield, as mentioned in the introduction of this paper. Thus, it should be important to study the molecular relationship between these traits, although there is no remarkable study so far. However, no study has been done on the relationship between these two traits at molecular level. In this paper, Liao et al. focused on two genes: SLR1, and MOC1, that is, SLR1 that functions to suppress gibberellin (GA) signaling resulting in dwarfism, and MOC1 that is a tiller regulating component. By morphological, biochemical, and molecular biological analyses, the authors revealed that SLR1 interacts with MOC1 and inhibits the degradation of MOC1, whereas GA causes stimulates the SLR1 degradation to thus bring about results in the inverse regulation of plant height and tiller number.

Response: Many thanks!

As mentioned, it is an indubitable fact that the integrated study on the molecular mechanism(s) controlling the tiller number and plant height should be important from the view point not only of in basic science but also of in practical real rice breeding. In this context, the studies by their study on revealing such molecular mechanism could be a epoch making pioneering work to directly reveal such molecular mechanism. However, this paper has many concerns, solitudes, and problematic issues for publication in a high impact journal. I listed my comments as follows.

Response: Thanks!

1. In Fig. 2a, I could not figure out that the length of axillary buds of *sd1* plant is longer than that of the control plant. Similarly, I can not exactly figure out the exact phenomenon phenotypes they mentioned about Fig. 2a (WT vs. *slr1*) or 2c (WT vs. SLR1-RNAi). In Sup. Fig. 3a, I realized that the right axillary bud at the right of the *slr1-d1* plant was longer than that of WT, but not in the case of the left side. Comparison Characterization of the length of axillary buds is a very important part for this paper, I would like to suggest the authors to present the state of axillary buds as like in Sup. Fig. 3c, by which readers could easily figure out the difference in the length of axillary buds.

Response: Sorry for the confusion. Because the first tiller bud of rice usually does not elongate, the length of the second tiller bud is used for comparison. In Fig. 2a and 2c, the second buds of *slr1* and SLR1-RNAi are much shorter than their wild-type. In the revised manuscript, we have added the number of buds with more clear descriptions.

2. In Fig. 2d, the authors did not mention about the SLR1 protein level, in Fig. 2d even though. In Fig. 2d, four bands with difference sizes were recognized by the SLR1 antibody. Among them, the authors suggested that the biggest and smallest bands correspond to SLR1-GFP and SLR1, but no without explanation for the other middle two bands in the

middle. Are these non-specific bands? Why were these intermediate bands were not detected in the SLR1 blot experiments for other plants in Fig. 2b or f? I would like to see the full-scans results of SLR1 blotting in Fig. 2b, 2f, Sup. Fig. 3b, and Sup. Fig. 3f, as like Fig. 2d.

Although figures in Sup. Fig. 9-14 are very good for correct discussion, After writing the above comment, I found the “full-scan”(the authors used the word, full-scan, but the photos presented in Sup. Fig. 9-14 are still closed view around the SLR1 bands in Sup. Fig. 9-14.

Although these figures are very good for correct discussion, they have problems, and . Actually, these are not real full-scan photos, but only closed up the neighborhood of SLR1 band, whereas the authors did not mention about these supplemental figures in the text at all. Furthermore, I could not recognize border lines between SLR1 and MOC1 blots in Sup. Fig. 9, 12, and 13, which makes me difficult to separate tow photos blotted by the SLR1 or MOC antibody.

Anyway, my above question is not settled by this full-scan result of Sup. Fig. 9b. On the contrary, after seeing these full-scan results, I had some other questions about the immunoactive bands. Actually, there are other immunoactive bands, which were not found in Fig. 2. The author just focused on the bands in the red box in Sup. Fig. 9 to present in Fig. 2. However, how did the authors recognize which band corresponds to the real SLR1/MOC1? In Sup. Fig. 9a, for example, there is one strong band showing higher mobility than the SLR1 band squared by as in the red box red (I am not sure whether the band with his higher mobility band really blotted by the SLR1 antibody), of the intensity of which the intensity is not different among the three samples. Then, how did the authors conclude that SLR1 really correspond to the bands squared by red line in the red box, but not to the lower bands? There are I have similar concerns for similar situations in other results not only in the case of SLR1 blot but in the results of MOC1. The authors should show a clear give a reason why the bands squared by a red line in the red box really correspond to the exact SLR1/MOC1 protein.

Response: Thanks. To prove that the bands are really corresponding to MOC1 proteins, we performed Western blots of MOC1 in the *moc1* mutant and its wild-type H89025, and the result showed that the MOC1 band disappeared in the mutant (new Supplementary Fig. 12b). Similar result is also obtained in the *moc1*^{CR} line generated by CRIPSR/Cas9 (new Supplementary Fig. 12c). To prove that the bands are really correspond to SLR1 proteins, we performed Western blots using anti-SLR1 and anti-GFP in the wild type, and SLR1-FLAG and SLR1-GFP protein extracts from rice protoplast. The result showed that anti-SLR1 could detect the SLR1-GFP protein (new Supplementary Fig. 12a). We have provided the full-scan results of Fig. 2b, 2d, 2f, 4g, 5a and Sup. Fig. 3b, 3f in new Supplementary Fig. 11 as suggested. The text is revised accordingly.

3. Among Fig. 2 and Sup. Fig. 3, the most important result could be Fig. 2e and f (Mock vs. GA- or PBZ-treatment) for their argument, I guess. Again, I could not see clear difference in the state of axillary buds or the MOC1 protein level among these plants (although I recognized delicate differences between Mock vs. GA- or PBZ-treatment). I would like to see clearer difference in the axillary bud length and clearer difference in protein blotting results of MOC1 and SLR1, as because these are the fundamental observations for their argument of in this paper.

Response: We have revised the manuscript to make it clearer and added the statistic data of the length of second axillary buds after the treatments of GA or PBZ in the new Supplement Fig. 4. We have repeated the protein blotting results of MOC1 and SLR1, which is consistent with previous result and updated in Fig. 2f.

4. About Sup. Fig. 4a, the authors mentioned that the numbers indicate the position of amino acids. I confirmed the correspondence between the numbers in Sup. Fig. 4a and the amino acid residue of rice MOC. The 61st amino acid of MOC1, which contains 441 amino acids, is Leu whereas, although this figure says N61. The 75th amino acid is Ile, while this figure says C75. Similarly, I found no correspondence between the amino acid position of SLR1 and the number problem is found in Sup. Fig. 5b: t. For example, the 230th amino acid of SLR1 is C, while the figure says N230, and the 169th is V but the figure says C169.

Response: Sorry for the confusion. We use 'MOC1N61' to represent the 61 amino acids at the N-terminal of the MOC1 protein. We now added a gene model in new Supplementary Fig. 6 and 7 to make it clear.

5. The authors mentioned that LHRI and VHIID motifs are important for interaction with MOC1. However the result in Sup. Fig. 4b shows that MOC1 Δ LHRI formed the strongest colony among the deletion series of SLR1, whereas MOC1 Δ VHIID formed the faintest one. Furthermore, although the authors mentioned that the LHRI motif of SLR1 is important for interaction with MOC1, I cannot find any difference in the colony formation between SLR1 Δ LHRI and other deletion series of SLR1 (Sup. Fig. 5c). The clearest effect was seen in SLR1N230 (located in polyS/T/V, but its position may be incorrect) and SLR1C75 (intervening region between DELLA and TVH) in Sup. Fig. 5b. I would like to strongly recommend that the authors redo a yeast two hybrid experiment for more quantitative way a better quantification by a liquid assay, not by colony assay.

Response: MOC1 and SLR1 both belong to the GRAS protein family, which contains intrinsically disordered protein structure (Dyson and Wright, 2005). This structure makes the interaction of GRAS proteins very complicated. Indeed, our truncation experiments showed a complicate result that none of single motifs is fully responsible for the interaction between SLR1 and MOC1. But for both MOC1 and SLR1, it is clear that truncation of the entire GRAS domain does abolish the interaction. We have revised the manuscript to make it clearer. As suggested, we also have repeatedly performed a liquid assay by measuring the β -galactosidase activity, however, it does not help a lot in answering this question (see below).

Figure: β -galactosidase activities detected in a liquid assay with Y187 transformants (mean \pm s.e.m., n = 3).

6. The results presented in Fig. 4a and b are very important observations for the conclusion of this work. As the authors have some other suitable plants for this experiment, such as *sd1*, *slr1*, *slr1-d1*, and GA20-1OE, I would like to see the results of the same experiment using these plants to confirm their conclusion. I wonder why they used another plant, *GID1OE*, for this experiment, but not plants such as *sd1*, *slr1*, *slr1-d1*, and GA20-1OE and not the above plants, which were already analyzed before to show a relationship between GA signaling and axially bud elongation. I would like to see the results for these plants in the same experiment.

Response: As suggested, we detected the degradation of MOC1 in *sd1*, which is consistent with other results. The new data is presented in new Supplementary Fig. 7a. We used *GID1OE* plants because the *slr1* mutant is sterile. Furthermore, we performed an additional experiment as shown in new Fig. 4c, showing that adding purified His-Trx-SLR1 proteins could inhibit the degradation of MOC1.

7. Based on the results in Fig. 4c-f and Sup. Fig. 7, the authors concluded that SLR1 inhibits the degradation of MOC1 independently of TAD1. I cannot agree with this conclusion only by these results, because the amount of SLR1 is higher in *tad1* than in WT (Fig. 4f), demonstrating that the *tad1* mutation causes higher accumulation of SLR1, and resulting in higher accumulation of MOC1. The authors should clarify the molecular/genetical relationship between TAD1 and SLR1 by other approaches.

Response: Our previous paper (Xu et al., Nat Commun, 2012) showed that MOC1 is the substrate of TAD1 for degradation. Here, we further showed that in the *tad1* mutant SLR1 protein could still inhibit the degradation of MOC1. We intend to draw the conclusion that SLR1 could inhibit the degradation of MOC1 without the function of TAD1, but not to exclude the indirect regulation by TAD1 through SLR1. Indeed, the relationship between TAD1 and SLR1 is unclear. We have revised it in the manuscript.

Response to the comments of Reviewer 2:

This is a well-written manuscript that presents novel data to explain the observation of increased tillering (branching) in gibberellin (GA)-deficient plants. Such plants accumulate DELLA protein and here it is shown that the rice DELLA, SLR1, interacts with and stabilises the tillering promoter MOC1. The interaction between SLR1 and MOC1 is demonstrated convincingly using different methods. It is also shown that MOC1 protein level is positively correlated with SLR1 content using numerous lines with altered SLR1 content and chemical treatments. Importantly, by crossing lines in which MOC1 and the GA-receptor GID1 are knocked down by CRISPR/Cas9, they show that GA-reduction of tiller number (but not its promotion of plant height) is dependent on MOC1 function.

Response: Many thanks!

These are important results of general interest. However, I have a question about the overall interpretation. The authors suggest that SLR1 promotes MOC1 function by stabilising it. However, the interaction of SLR1 with MOC1 would be expected to affect its function.

Another interpretation is that SLR1 and MOC1 act as a complex to promote branching. There are many examples of DELLA proteins acting as transcriptional co-activators in combination with transcription factors. I don't believe that the results presented in this manuscript discount this possibility.

Response: Thanks, we have also been curious about the transcriptional regulation of MOC1. However, we found MOC1 has no transcriptional activity in the luciferase assay, and SLR1 could not change the transcriptional activity of MOC1. As no target gene of MOC1 has been identified so far yet, we lack an experimental way to prove it or discount this possibility.

One aspect of the work I did not find convincing was the yeast 2-hybrid assays to determine the protein motifs in MOC1 and SLR1 that are necessary for the interaction. Clearly a lot of work was put into this, but the results seem unclear. For example, the authors suggested that the LHR1 motif in MOC1 is important for the interaction with SLR1, yet in Fig S4, the Δ LHR1 sequence appears to produce one of the strongest interactions. I think the statement on page 9 that LHR1 and VHIID motifs of MOC1 are essential for the interaction should be reconsidered. In Figs S4 and S5, it is unclear what the C-terminal fragments refer to. Does the number in for example MOC1C75 refer to the numbers of residues missing from the C terminus or to a fragment containing amino acids from 75 to the C terminus?

Response: Sorry for the confusion. We use 'MOC1N61' to represent the 61 amino acids at the N-terminal of the MOC1 protein. We now added a gene model in new Supplementary Fig. 6 and 7 to make it clear. We have revised the manuscript as suggested.

It would be helpful to the reader, particularly to one not familiar with the GA-signalling field, if the mutants used in this study are explained. For example, it should be explained that the slr1 recessive mutant results in loss of function due to mutation in the GRAS domain while the slr1-d1 dominant mutant causes stabilisation and gain of function due to an N-terminal mutation. It could also be pointed out that the sd1 mutant is GA-deficient due to knock-down or loss of GA20ox2. Fig S1 could be helpful for this purpose.

Response: Thanks. We have added the information in the revised manuscript.

Response to the comments of Reviewer 3:

In rice, tiller number and plant height are usually inversely regulated. Increase in GA levels promotes plant height and reduces tiller number, whereas impaired GA signaling or biosynthesis reduces plant height and increases tiller number. SLR1 is known to suppress plant height and MOC1 to promote tiller number. Authors of this manuscript explored the mechanism of tiller number suppression and plant height promotion by GA. They found that SLR1 interacts with MOC1 and inhibits the degradation of MOC1. With increased levels of GA, levels of both SLR1 and MOC1 were shown to be slightly reduced and proposed as a mechanism for increase in plant height and decrease in tiller number. Various rice mutants and GA and PBZ (GA biosynthesis inhibitor) treatments were employed in all experiments. The pathway of GA positively regulating GID1 (GA receptor), which negatively regulating

SLR1, which positively regulating MOC1 seems to be supported by some data. However, the role of TAD1 in the regulation of MOC1 and SLR1 is not completely supported by data. Specific comments are described as follows:

Response: Many thanks.

1) Fig. 2: The increase in both SLR1 and MOC1 levels in *sd1* (the GA receptor) mutant and decrease in both SLR1 and MOC1 levels in *slr1* mutant (Fig. 2b) do not seem to be as significantly as treatment of WT with PBZ and GA (Fig. 2f), suggesting that degradation of the SLR1-MOC1 complex by GA may involve other factors.

Response: Thanks and we have added this in the discussion.

2) Fig. 4: The E coli-expressed MBP-MOC1 fusion protein was incubated with cell extracts from SLR1-GFP and GID1-overexpression transgenic plants (Fig. 4a, b). MBP-MOC1 levels were shown to be stabilized in SLR1-GFP-overexpressing extracts but degraded rapidly in GID1-overexpressing extracts. However, the levels of overexpressed SLR1-GFP and GID1 in cell extracts throughout the incubation time were not provided, making it uncertain whether the stabilization or degradation of MBP-MOC1 was indeed caused by overexpression of SLR1-GFP and GID1. Cell extract from GFP-overexpressing transgenic plants should also be provided as a control. It is possible that non-specific protein degradation by general proteases may occur, and overexpression of certain proteins may dilute out these proteases present in cell extracts.

Response: We performed an additional experiment to address this question as shown in new Fig. 4c, which showed that adding the purified His-Trx-SLR1 protein could inhibit the degradation of MOC1, demonstrating that the inhibition of MOC1 degradation is caused by SLR1 overexpression.

Since GA promotes the degradation of SLR1 via the 26S proteasome pathway, GID1-overexpression should increase the degradation of SLR1. Whether the 26S proteasome inhibitor MG132 can stabilize the MBP-MOC1 complex should be tested to demonstrate the specificity of MOC1 degradation by the GID1-mediated 26S proteasome pathway.

Response: As suggested, we added MG132 in the vitro cell-free protein degradation assay and found that MG132 indeed could stabilize the MBP-MOC1. The data is shown in new Supplementary Fig. 7b.

The pathway proposed in Fig. 5j shows that TAD1 interacts and degrades MOC1, which acts independently of the GA signaling pathway. However, in the *tad1* mutant, SLR1 level was increased (Fig. 4f) and plant height was reduced as compared with the WT (Fig. 4e), indicating TAD1 suppressed SLR1 level. Additionally, treatment of *tad1* mutant with GA reduced SLR1 level by 2-fold (Fig. 4f) but still reduced plant height (Fig. 4e). These results indicate that TAD1 may crosstalk with the GA signaling pathway in regulating plant height, and SLR1 may not be the only negative regulator of plant height. The notion is supported by data in Supplementary Fig 6b,c, in which tiller number is increased and plant height is

reduced in *tad1* mutant, but both not in *tad1/GID1-OE*. The notion is also supported by some data in Supplementary Fig. 7a, b.

Response: We agreed with the reviewer and discussed more in the revised manuscript.

3) Fig. 5: Since the interaction between SLR1 and MOC1 stabilizes the two proteins, why the accumulation of MOC1 is abolished but that of SLR1 remains the same in the *moc1/crispr* mutant as compared with the WT (Fig. 5a)?

Response: The interaction between SLR1 and MOC1 stabilizes MOC1 but does not influence the stability of SLR1, as shown in Supplementary Fig. 10e.

4) In summary, some data are contradictory and inconsistent as mentioned above, and the proposed model in Fig. 5 j may need modification.

Response: Thanks. We have revised Fig. 5j as suggested.

Reviewers' comments:

Reviewer #1 (Remarks to the Author):

I found that the authors succeeded to collectively respond to my questions and concerns in the revised version. I believe that this paper provides a new concept on molecular relationship between MOC1 and SLR1, and consequently on grain yield and plant architecture.

1. Sorry for the confusion. Because the first tiller bud of rice usually does not elongate, the length of the second tiller bud is used for comparison. In Fig. 2a and 2c, the second buds of *slr1* and SLR1-RNAi are much shorter than their wild-type. In the revised manuscript, we have added the number of buds with more clear descriptions.

Now I can discern the bud length difference in these mutants and corresponding WT.

2. To prove that the bands are really corresponding to MOC1 proteins, we performed Western blots of MOC1 in the *moc1* mutant and its wild-type H89025, and the result showed that the MOC1 band disappeared in the mutant (new Supplementary Fig. 12b). Similar result is also obtained in the *moc1CR* line generated by CRISPR/Cas9 (new Supplementary Fig. 12c). To prove that the bands are really correspond to SLR1 proteins, we performed Western blots using anti-SLR1 and anti-GFP in the wild type, and SLR1-FLAG and SLR1-GFP protein extracts from rice protoplast. The result showed that anti-SLR1 could detect the SLR1-GFP protein (new Supplementary Fig. 12a). We have provided the full-scan results of Fig. 2b, 2d, 2f, 4g, 5a and Sup. Fig. 3b, 3f in new Supplementary Fig. 11 as suggested. The text is revised accordingly.

Actually, the antibody specificity is very important for this study. The added results in the revised version should be trustable to sweep away my concern about this issue.

3. We have revised the manuscript to make it clearer and added the statistic data of the length of second axillary buds after the treatments of GA or PBZ in the new Supplement Fig. 4. We have repeated the protein blotting results of MOC1 and SLR1, which is consistent with previous result and updated in Fig2f.

The new experimental results are accord with my expectation.

4. Sorry for the confusion. We use „MOC1N61“ to represent the 61 amino acids at the N-terminal of the MOC1 protein. We now added a gene model in new Supplementary Fig.6 and 7 to make it clear.

My concern was swept away by the above explanation.

5. MOC1 and SLR1 both belong to the GRAS protein family, which Contains intrinsically disordered protein structure (Dyson and Wright, 2005). This structure makes the interaction of GRAS proteins very complicated. Indeed, our truncation experiments showed a complicate result that none of single motifs is fully responsible for the interaction between SLR1 and MOC1. But for both MOC1 and SLR1, it is clear that truncation of the entire GRAS domain does abolish the interaction. We have revised the manuscript to make it clearer. As suggested, we also have repeatedly performed a liquid assay by measuring the β -galactosidase activity, however, it does not help a lot in answering this question (see below).

Accurate evaluation interaction motif may be helpful for understanding how two proteins separately involve in plant height and tillers formation. Nevertheless, the present results are enough to prove the protein interaction.

6. As suggested, we detected the degradation of MOC1 in *sd1*, which is consistent with other results. The new data is presented in new Supplementary Fig. 7a. We used GID1OE plants because the *slr1* mutant is sterile. Furthermore, we performed an additional experiment as shown in new Fig. 4c, showing that adding purified His-Trx-SLR1 proteins could inhibit the degradation of MOC1.

After thorough in vitro cell-free protein degradation assay in GA- related mutant plant tissue, such as *sd1* and *GID1OE*, the part of results turned to be more solid than before which demonstrate the degradation of MOC1 affected by GA signaling.

7. Our previous paper (Xu et al., Nat Commun, 2012) showed that MOC1 is the substrate of TAD1 for degradation. Here, we further showed that in the *tad1* mutant SLR1 protein could still inhibit the degradation of MOC1. We intend to draw the conclusion that SLR1 could inhibit the degradation of MOC1 without the function of TAD1, but not to exclude the indirect regulation by TAD1 through SLR1. Indeed, the relationship between TAD1 and SLR1 is unclear. We have revised it in the manuscript.

Based on phenotypical analysis of *tad1*, *tad1SLR-RNAi* and *tad1GIDOE*, the explanation is getting reasonable. The actual molecular relationship between TAD1 and SLR1 needs to be clarified in the future.

Reviewer #3 (Remarks to the Author):

Reviewer 3, comment to the response:

1)

Answer to this point could not be found in the Discussion section.

2)

1. The additional experiment in Fig. 4c did not address questions directly.

2. In the new Fig. 4c, why His-TRx-SLR1 protein disappeared 30 min after incubation with MBP-MOC1?

3. The kinetics of MBP-MOC1 degradation in WT is quite different between Fig. 4a/4b and Fig. 4c, and between Supplementary Fig. 7a and Fig. 7b. In fact, the half-life of MBP-MOC1 in WT could be even shorter than in The *GIDOE* extract based on the band density in Supplementary Fig. 7b.

Although the addition of MG132 could slow down the degradation of MBP-MOC1, it did not inhibit the degradation of MBP-MOC1 as the authors claimed (lines 175-177). Therefore, the conclusion is not fully supported by the experiment of Supplementary Fig. 7b.

3)

The author claimed that SLR1 stabilize MOC, but MOC does not affect the stability of SLR1. If this is the case, the level of MOC1 should be higher and lower, when that of SLR1 is higher and lower, respectively. However, the relative abundance of SLR1 compared to MOC1 is opposite in two WTs in Supplementary Fig. 3d and Fig. 3f, suggesting that the abundance of SLR and MOC1 are correlated, but it is necessary that SLR stabilizes MOC1. The effect on the stability or half-life of MOC1 by SLR1 should be determined under the conditions of inhibition of transcription (such as using actinomycin D) and translation (such as using cycloheximide) experiments.

Point-by-point response to Reviewers

Response to the comments of Reviewer 1:

I found that the authors succeeded to collectively respond to my questions and concerns in the revised version. I believe that this paper provides a new concept on molecular relationship between MOC1 and SLR1, and consequently on grain yield and plant architecture.

Response: Many thanks!

1. Sorry for the confusion. Because the first tiller bud of rice usually does not elongate, the length of the second tiller bud is used for comparison. In Fig. 2a and 2c, the second buds of *slr1* and SLR1-RNAi are much shorter than their wild-type. In the revised manuscript, we have added the number of buds with more clear descriptions.

Now I can discern the bud length difference in these mutants and corresponding WT.

Response: Thanks!

2. To prove that the bands are really corresponding to MOC1 proteins, we performed Western blots of MOC1 in the *moc1* mutant and its wild-type H89025, and the result showed that the MOC1 band disappeared in the mutant (new Supplementary Fig. 12b). Similar result is also obtained in the *moc1CR* line generated by CRIPSR/Cas9 (new Supplementary Fig. 12c). To prove that the bands are really correspond to SLR1 proteins, we performed Western blots using anti-SLR1 and anti-GFP in the wild type, and SLR1-FLAG and SLR1-GFP protein extracts from rice protoplast. The result showed that anti-SLR1 could detect the SLR1-GFP protein (new Supplementary Fig. 12a). We have provided the full-scan results of Fig. 2b, 2d, 2f, 4g, 5a and Sup. Fig. 3b, 3f in new Supplementary Fig. 11 as suggested. The text is revised accordingly.

Actually, the antibody specificity is very important for this study. The added results in the revised version should be trustable to sweep away my concern about this issue.

Response: Many thanks!

3. We have revised the manuscript to make it clearer and added the statistic data of the length of second axillary buds after the treatments of GA or PBZ in the new Supplement Fig. 4. We have repeated the protein blotting results of MOC1 and SLR1, which is consistent with previous result and updated in Fig2f.

The new experimental results are accord with my expectation.

Response: Thanks!

4. Sorry for the confusion. We use „MOC1N61“ to represent the 61 amino acids at the N-terminal of the MOC1 protein. We now added a gene model in new Supplementary Fig.6 and 7 to make it clear.

My concern was swept away by the above explanation.

Response: Thanks!

5. MOC1 and SLR1 both belong to the GRAS protein family, which contains intrinsically disordered protein structure (Dyson and Wright, 2005). This structure makes the interaction of GRAS proteins very complicated. Indeed, our truncation experiments showed a complicated result that none of single motifs is fully responsible for the interaction between SLR1 and MOC1. But for both MOC1 and SLR1, it is clear that truncation of the entire GRAS domain does abolish the interaction. We have revised the manuscript to make it clearer. As suggested, we also have repeatedly performed a liquid assay by measuring the β -galactosidase activity, however, it does not help a lot in answering this question (see below).

Accurate evaluation interaction motif may be helpful for understanding how two proteins separately involve in plant height and tillers formation. Nevertheless, the present results are enough to prove the protein interaction.

Response: Thanks!

6. As suggested, we detected the degradation of MOC1 in *sd1*, which is consistent with other results. The new data is presented in new Supplementary Fig. 7a. We used *GID1OE* plants because the *slr1* mutant is sterile. Furthermore, we performed an additional experiment as shown in new Fig. 4c, showing that adding purified His-Trx-SLR1 proteins could inhibit the degradation of MOC1.

After thorough *in vitro* cell-free protein degradation assay in GA-related mutant plant tissue, such as *sd1* and *GID1OE*, the part of results turned to be more solid than before which demonstrate the degradation of MOC1 affected by GA signaling.

Response: Many thanks!

7. Our previous paper (Xu et al., Nat Commun, 2012) showed that MOC1 is the substrate of TAD1 for degradation. Here, we further showed that in the *tad1* mutant SLR1 protein could still inhibit the degradation of MOC1. We intend to draw the conclusion that SLR1 could inhibit the degradation of MOC1 without the function of TAD1, but not to exclude the indirect regulation by TAD1 through SLR1. Indeed, the relationship between TAD1 and SLR1 is unclear. We have revised it in the manuscript.

Based on phenotypical analysis of *tad1*, *tad1SLR-RNAi* and *tad1GIDOE*, the explanation is getting reasonable. The actual molecular relationship between TAD1 and SLR1 needs to be clarified in the future.

Response: Thanks!

Response to the comments of Reviewer 3:

1)

Answer to this point could not be found in the Discussion section.

Response: Thanks for your comments. We add more discussion in the revised manuscript (see highlight in line 262-265 at page 10).

2)

1. The additional experiment in Fig. 4c did not address questions directly.

Response: Actually this result showed that addition of the purified His-Trx-SLR1 protein in the cell-free degradation system can dramatically inhibit the degradation of MOC1 but His-Trx-GST cannot, proving that the stabilization of MOC1 was indeed caused by SLR1.

2. In the new Fig. 4c, why His-TRx-SLR1 protein disappeared 30 min after incubation with MBP-MOC1?

Response: It is because SLR1 degrades fast in rice (Fig. 4c and Supplementary Fig. 10e). Previous study showed that all five *Arabidopsis* DELLA proteins including GA INSENSITIVE (GAI), REPRESSOR OF ga1-3 (RGA), and three REPRESSOR OF ga1-3-LIKE proteins (RGL1, RGL2, and RGL3) are degraded within 30 minutes (Figure 1 in Wang et al, Plant Cell, 2009; 21:2378-2390). We added this information in the revised manuscript (see highlight in line 205-208 at page 8).

Figure 1. Cell-Free Degradation of Plant-Derived DELLA Proteins.

(A) and (B) Cell-free degradation of TAP-tagged RGA (A) and GAI (B) proteins. Protein extracts were prepared from 7-d-old 35S:TAP-RGA/rga-24 and 35S:TAP-GAI/gai-t6 seedlings and then incubated with or without MG132 over the indicated time course. RPN6 was used as a nondegraded loading control. Start: time point zero in each degradation assay.

(C) and (D) Half-life plot for cell-free degradation of TAP-RGA and TAP-GAI in (A) and (B), respectively.

(E) Cell-free degradation of all five DELLA proteins. Wild-type plants expressing 35S promoter-driven full-length RGA, GAI, RGL1, RGL2, and RGL3 transgenes tagged with TAP at the N terminus were used to perform the degradation assays.

(F) Degradation of TAP-RGA in wild-type extracts. TAP-RGA was enriched from 7-d-old 35S:TAP-RGA/rga-24 seedlings by affinity matrix and then incubated in wild-type extracts with or without MG132.

(G) Half-life plot for cell-free degradation of TAP-RGA in (F).

3. The kinetics of MBP-MOC1 degradation in WT is quite different between Fig. 4a/Fig. 4b and Fig. 4c, and between Supplementary Fig. 7a and Fig. 7b. In fact, the half-life of MBP-MOC1 in WT could be even shorter than in The GIDOE extract based on the band density in Supplementary Fig. 7b.

Response: The kinetics of MBP-MOC1 degradation could be influenced by the growth conditions of plants. For each experiment, to ensure that all the plants grew under the same condition, we grew them together. Importantly, the conclusions are consistent among all the experiments.

Although the addition of MG132 could slow down the degradation of MBP-MOC1, it did not inhibit the degradation of MBP-MOC1 as the authors claimed (lines 175-177). Therefore, the conclusion is not fully supported by the experiment of Supplementary Fig. 7b.

Response: Many thanks! We have revised the manuscript to draw an accurate conclusion (see highlight in line 179-181 at page 7).

3)

The author claimed that SLR1 stabilize MOC, but MOC does not affect the stability of SLR1. If this is the case, the level of MOC1 should be higher and lower, when that of SLR1 is higher and lower, respectively. However, the relative abundance of SLR1 compared to MOC1 is opposite in two WT's in Supplementary Fig. 3d and Fig. 3f, suggesting that the abundance of SLR and MOC1 are correlated, but it is necessary that SLR stabilizes MOC1. The effect on the stability or half-life of MOC1 by SLR1 should be determined under the conditions of inhibition of transcription (such as using actinomycin D) and translation (such as using cycloheximide) experiments.

Response: MOC1 protein levels are increased in both *slr1-d1* and *d18*, and the increase fold is higher in *d18*. This is consistent with the observation that *d18* exhibits longer axillary buds than *slr1-d1* (Supplementary Fig. 3). It is possible that other factors in the downstream of D18 affect MOC1. Moreover, our results showed the MOC1 protein degraded rapidly within 30 minutes and addition of purified His-Trx-SLR1 protein in the cell-free degradation system can dramatically inhibit the degradation of MOC1 but not the His-Trx-GST protein (Fig. 4c). As the secondary transcriptional effect usually happened more than 30 minutes, it is rational to assume that the stabilization of MOC1 was directly caused by SLR1. We have revised the result and added some discussion in the revised manuscript (see highlight in line 173-179 at page 7 and line 260-265 at page 10).

Point-by-point response to Editor and Reviewers

Response to the comments of Reviewer 1:

I found that the authors succeeded to collectively respond to my questions and concerns in the revised version. I believe that this paper provides a new concept on molecular relationship between MOC1 and SLR1, and consequently on grain yield and plant architecture.

Response: Many thanks!

1. Sorry for the confusion. Because the first tiller bud of rice usually does not elongate, the length of the second tiller bud is used for comparison. In Fig. 2a and 2c, the second buds of *slr1* and SLR1-RNAi are much shorter than their wild-type. In the revised manuscript, we have added the number of buds with more clear descriptions.

Now I can discern the bud length difference in these mutants and corresponding WT.

Response: Thanks!

2. To prove that the bands are really corresponding to MOC1 proteins, we performed Western blots of MOC1 in the *moc1* mutant and its wild-type H89025, and the result showed that the MOC1 band disappeared in the mutant (new Supplementary Fig. 12b). Similar result is also obtained in the *moc1CR*

line generated by CRIPSR/Cas9 (new Supplementary Fig. 12c). To prove that the bands are really correspond to SLR1 proteins, we performed Western blots using anti-SLR1 and anti-GFP in the wild type, and SLR1-FLAG and SLR1-GFP protein extracts from rice protoplast. The result showed that anti-SLR1 could detect the SLR1-GFP protein (new Supplementary Fig. 12a). We have provided the full-scan results of Fig. 2b, 2d, 2f, 4g, 5a and Sup. Fig. 3b, 3f in new Supplementary Fig. 11 as suggested. The text is revised accordingly.

Actually, the antibody specificity is very important for this study. The added results in the revised version should be trustable to sweep away my concern about this issue.

Response: Many thanks!

3. We have revised the manuscript to make it clearer and added the statistic data of the length of second axillary buds after the treatments of GA or PBZ in the new Supplement Fig. 4. We have repeated the protein blotting results of MOC1 and SLR1, which is consistent with previous result and updated in Fig2f.

The new experimental results are accord with my expectation.

Response: Thanks!

4. Sorry for the confusion. We use „MOC1N61“ to represent the 61 amino acids at the N-terminal of the MOC1 protein. We now added a gene model in new Supplementary Fig.6 and 7 to make it clear.

My concern was swept away by the above explanation.

Response: Thanks!

5. MOC1 and SLR1 both belong to the GRAS protein family, which Contains intrinsically disordered protein structure (Dyson and Wright, 2005). This structure makes the interaction of GRAS proteins very complicated. Indeed, our truncation experiments showed a complicate result that none of single motifs is fully responsible for the interaction between SLR1 and MOC1. But for both MOC1 and SLR1, it is clear that truncation of the entire GRAS domain does abolish the interaction. We have revised the manuscript to make it clearer. As suggested, we also have repeatedly performed a liquid assay by measuring the β -galactosidase activity, however, it does not help a lot in answering this question (see below).

Accurate evaluation interaction motif may be helpful for understanding how two proteins separately involve in plant height and tillers formation. Nevertheless, the present results are enough to prove the protein interaction.

Response: Thanks!

6. As suggested, we detected the degradation of MOC1 in sd1, which is consistent with other results. The new data is presented in new Supplementary Fig. 7a. We used GID1OE plants because the slr1 mutant is sterile. Furthermore, we performed an additional experiment as shown in new Fig. 4c, showing that adding purified His-Trx-SLR1 proteins could inhibit the degradation of MOC1.

After thorough in vitro cell-free protein degradation assay in GA- related mutant plant tissue, such as *sd1* and *GID1OE*, the part of results turned to be more solid than before which demonstrate the degradation of MOC1 affected by GA signaling.

Response: Many thanks!

7. Our previous paper (Xu et al., Nat Commun, 2012) showed that MOC1 is the substrate of TAD1 for degradation. Here, we further showed that in the *tad1* mutant SLR1 protein could still inhibit the degradation of MOC1. We intend to draw the conclusion that SLR1 could inhibit the degradation of MOC1 without the function of TAD1, but not to exclude the indirect regulation by TAD1 through SLR1. Indeed, the relationship between TAD1 and SLR1 is unclear. We have revised it in the manuscript.

Based on phenotypical analysis of *tad1*, *tad1SLR-RNAi* and *tad1GIDOE*, the explanation is getting reasonable. The actual molecular relationship between TAD1 and SLR1 needs to be clarified in the future.

Response: Thanks!

Response to the comments of Reviewer 3:

1)

Fig. 2: The increase in both SLR1 and MOC1 levels in *sd1* (the GA receptor) mutant and decrease in both SLR1 and MOC1 levels in *slr1* mutant (Fig. 2b) do not seem to be as significantly as treatment of WT with PBZ and GA (Fig. 2f), suggesting that degradation of the SLR1-MOC1 complex by GA may involve other factors.

Answer to this point could not be found in the Discussion section.

Response: Thanks for your comments. We add more discussion in the revised manuscript (see highlight at page 10).

2)

1. The additional experiment in Fig. 4c did not address questions directly.

Response: We have repeated the experiment with *GFPOE* plants as the control. The result showed that MOC1 degradation was significantly reduced in the extracts from *SLR1-GFPOE* plants but not in *GFPOE* plants. The new result is shown in new Fig. 4a (Line 172-175 at page 7) in the revised manuscript.

2. In the new Fig. 4c, why His-TRx-SLR1 protein disappeared 30 min after incubation with MBP-MOC1?

Response: Previous study showed that all five *Arabidopsis* DELLA proteins are degraded within 30 minutes (Wang et al, Plant Cell, 2009; 21:2378-2390). We also added this information in the revised manuscript (see highlight at page 8).

We have repeated the experiment using longer time (up to 180 min) with three biological replicates. Hix-Trx-SLR1 indeed degraded rapidly within 60 min in rice. Addition of Hix-Trx-SLR1 significantly delayed the degradation of MOC1, as MOC1 almost degraded at 60 min without Hix-Trx-SLR1 and

degraded at 120 min with the addition of Hix-Trx-SLR1. The new results are shown in new Fig. 4c and Supplemental Fig. 8 in the revised manuscript.

3. The kinetics of MBP-MOC1 degradation in WT is quite different between Fig. 4a/4b and Fig. 4c, and between Supplementary Fig. 7a and Fig. 7b. In fact, the half-life of MBP-MOC1 in WT could be even shorter than in The GIDOE extract based on the band density in Supplementary Fig. 7b.

Response: The kinetics of MBP-MOC1 degradation could be influenced by the growth conditions of plants. For each experiment, to ensure that all the plants grew under the same condition, we grew them together. Importantly, the conclusions are consistent among all the experiments.

Although the addition of MG132 could slow down the degradation of MBP-MOC1, it did not inhibit the degradation of MBP-MOC1 as the authors claimed (lines 175-177). Therefore, the conclusion is not fully supported by the experiment of Supplementary Fig. 7b.

Response: Many thanks! We have revised the manuscript to draw an accurate conclusion (see highlight in Line 181 at page 7).

3)

The author claimed that SLR1 stabilize MOC, but MOC does not affect the stability of SLR1. If this is the case, the level of MOC1 should be higher and lower, when that of SLR1 is higher and lower, respectively. However, the relative abundance of SLR1 compared to MOC1 is opposite in two WTs in Supplementary Fig. 3d and Fig. 3f, suggesting that the abundance of SLR and MOC1 are correlated, but it is necessary that SLR stabilizes MOC1. The effect on the stability or half-life of MOC1 by SLR1 should be determined under the conditions of inhibition of transcription (such as using actinomycin D) and translation (such as using cycloheximide) experiments.

Response: MOC1 protein levels are increased in both *slr1-d1* and *d18*, and the increase fold is higher in *d18*. This is consistent with the observation that *d18* exhibits longer axillary buds than *slr1-d1* (Supplementary Fig. 3). It is possible that other factors in the downstream of D18 affect MOC1. Moreover, our results showed the MOC1 protein degraded rapidly within 30 minutes and addition of purified His-Trx-SLR1 protein in the cell-free degradation system can dramatically inhibit the degradation of MOC1 but not the His-Trx-GST protein (Fig. 4c). As the secondary transcriptional effect usually happened more than 30 minutes, it is rational to assume that the stabilization of MOC1 was directly caused by SLR1. We have updated the result and added some discussion in the revised manuscript (see highlights, Line 172-179 at page 7 and page 10).

REVIEWERS' COMMENTS:

Reviewer #3 (Remarks to the Author):

Response to the comments of Reviewer 3:

General comment: The English language should be edited by an English editor.

1)

Fig. 2: The increase in both SLR1 and MOC1 levels in *sd1* (the GA receptor) mutant and decrease in both SLR1 and MOC1 levels in *slr1* mutant (Fig. 2b) do not seem to be as significantly as treatment of WT with PBZ and GA (Fig. 2f), suggesting that degradation of the SLR1-MOC1 complex by GA may involve other factors.

Answer to this point could not be found in the Discussion section.

Response: Thanks for your comments. We add more discussion in the revised manuscript (see highlight at page 10).

Comment: This is OK now.

2)

1. The additional experiment in Fig. 4c did not address questions directly.

Response: We have repeated the experiment with GFPOE plants as the control. The result showed that MOC1 degradation was significantly reduced in the extracts from SLR1-GFPOE plants but not in GFPOE plants. The new result is shown in new Fig. 4a (Line 172-175 at page 7) in the revised manuscript.

Comment: The repeated experiments with controls are better now.

2. In the new Fig. 4c, why His-TRx-SLR1 protein disappeared 30 min after incubation with MBP-MOC1?

Response: Previous study showed that all five Arabidopsis DELLA proteins are degraded within 30 minutes (Wang et al, Plant Cell, 2009; 21:2378-2390). We also added this information in the revised manuscript (see highlight at page 8).

We have repeated the experiment using longer time (up to 180 min) with three biological replicates.

Hix-Trx-SLR1 indeed degraded rapidly within 60 min in rice. Addition of Hix-Trx-SLR1 significantly delayed the degradation of MOC1, as MOC1 almost degraded at 60 min without Hix-Trx-SLR1 and degraded at 120 min with the addition of Hix-Trx-SLR1. The new results are shown in new Fig. 4c and

Supplemental Fig. 8 in the revised manuscript.

Comment: The repeated experiments are better now. The degradation of MBP-MOC1 is slowdown by incubation with His-Trx-SLR1. However, MBP-MOC1 doesn't seem to be significantly stabilized by SLR1. The *in vivo* interaction of native MOC1 and SLR1 should be further proved in the future. This point should be raised in the discussion.

3. The kinetics of MBP-MOC1 degradation in WT is quite different between Fig. 4a/Fig. 4b and Fig. 4c, and between Supplementary Fig. 7a and Fig. 7b. In fact, the half-life of MBP-MOC1 in WT could

be even shorter than in The GIDOE extract based on the band density in Supplementary Fig. 7b.

Response: The kinetics of MBP-MOC1 degradation could be influenced by the growth conditions of

plants. For each experiment, to ensure that all the plants grew under the same condition, we grew them together. Importantly, the conclusions are consistent among all the experiments.

Comment: OK.

Although the addition of MG132 could slow down the degradation of MBP-MOC1, it did not inhibit the degradation of MBP-MOC1 as the authors claimed (lines 175-177). Therefore, the conclusion is not fully supported by the experiment of Supplementary Fig. 7b.

Response: Many thanks! We have revised the manuscript to draw an accurate conclusion (see highlight in Line 181 at page 7).

Comment: OK.

3)

The author claimed that SLR1 stabilize MOC, but MOC does not affect the stability of SLR1. If this is the case, the level of MOC1 should be higher and lower, when that of SLR1 is higher and lower, respectively. However, the relative abundance of SLR1 compared to MOC1 is opposite in two WTs in

Supplementary Fig. 3d and Fig. 3f, suggesting that the abundance of SLR and MOC1 are correlated,

but it is necessary that SLR stabilizes MOC1. The effect on the stability or half-life of MOC1 by SLR1

should be determined under the conditions of inhibition of transcription (such as using actinomycin D)

and translation (such as using cycloheximide) experiments.

Response: MOC1 protein levels are increased in both slr1-d1 and d18, and the increase fold is higher

in d18. This is consistent with the observation that d18 exhibits longer axillary buds than slr1-d1 (Supplementary Fig. 3). It is possible that other factors in the downstream of D18 affect MOC1.

Moreover, our results showed the MOC1 protein degraded rapidly within 30 minutes and addition of

purified His-Trx-SLR1 protein in the cell-free degradation system can dramatically inhibit the degradation of MOC1 but not the His-Trx-GST protein (Fig. 4c). As the secondary transcriptional effect usually happened more than 30 minutes, it is rational to assume that the stabilization of MOC1

was directly caused by SLR1. We have updated the result and added some discussion in the revised

manuscript (see highlights, Line 172-179 at page 7 and page 10).

Comment: OK.

Point by point response to reviewer's comments

Reviewer #3

General comment: The English language should be edited by an English editor.

Response: Thanks!

1)

Fig. 2: The increase in both SLR1 and MOC1 levels in *sd1* (the GA receptor) mutant and decrease in both SLR1 and MOC1 levels in *slr1* mutant (Fig. 2b) do not seem to be as significantly as treatment of WT with PBZ and GA (Fig. 2f), suggesting that degradation of the SLR1-MOC1 complex by GA may involve other factors.

Answer to this point could not be found in the Discussion section.

Response: Thanks for your comments. We add more discussion in the revised manuscript (see highlight at page 10).

Comment: This is OK now.

Response: Thanks!

2)

1. The additional experiment in Fig. 4c did not address questions directly.

Response: We have repeated the experiment with GFPOE plants as the control. The result showed that MOC1 degradation was significantly reduced in the extracts from SLR1-GFPOE plants but not in GFPOE plants. The new result is shown in new Fig. 4a (Line 172-175 at page 7) in the revised manuscript.

Comment: The repeated experiments with controls are better now.

Response: Thanks!

2. In the new Fig. 4c, why His-TRx-SLR1 protein disappeared 30 min after incubation with MBP-MOC1?

Response: Previous study showed that all five Arabidopsis DELLA proteins are degraded within 30

minutes (Wang et al, Plant Cell, 2009; 21:2378-2390). We also added this information in the

revised manuscript (see highlight at page 8). We have repeated the experiment using longer time (up to 180 min) with three biological replicates. Hix-Trx-SLR1 indeed degraded rapidly within 60 min in rice. Addition of Hix-Trx-SLR1 significantly delayed the degradation of MOC1, as MOC1 almost degraded at 60 min without Hix-Trx-SLR1 and degraded at 120 min with the addition of Hix-Trx-SLR1. The new results are shown in new Fig. 4c and Supplemental Fig. 8 in the revised manuscript.

Comment: The repeated experiments are better now. The degradation of MBP-MOC1 is slowdown by incubation with His-Trx-SLR1. However, MBP-MOC1 doesn't seem to be significantly stabilized by SLR1. The in vivo interaction of native MOC1 and SLR1 should be further proved in the future. This point should be raised in the discussion.

Response: Thanks! We have added discussion about this in the revised manuscript.

3. The kinetics of MBP-MOC1 degradation in WT is quite different between Fig. 4a/Fig. 4b and Fig.

4c, and between Supplementary Fig. 7a and Fig. 7b. In fact, the half-life of MBP-MOC1 in WT could be even shorter than in The GIDOE extract based on the band density in Supplementary Fig. 7b.

Response: The kinetics of MBP-MOC1 degradation could be influenced by the growth conditions of plants. For each experiment, to ensure that all the plants grew under the same condition, we grew them together. Importantly, the conclusions are consistent among all the experiments.

Comment: OK.

Response: Thanks!

Although the addition of MG132 could slow down the degradation of MBP-MOC1, it did not inhibit the degradation of MBP-MOC1 as the authors claimed (lines 175-177). Therefore, the conclusion is not fully supported by the experiment of Supplementary Fig. 7b.

Response: Many thanks! We have revised the manuscript to draw an accurate conclusion (see highlight in Line 181 at page 7).

Comment: OK.

Response: Thanks!

3)

The author claimed that SLR1 stabilize MOC, but MOC does not affect the stability of SLR1. If this is the case, the level of MOC1 should be higher and lower, when that of SLR1 is higher and lower, respectively. However, the relative abundance of SLR1 compared to MOC1 is opposite in two WT in Supplementary Fig. 3d and Fig. 3f, suggesting that the abundance of SLR and MOC1 are correlated, but it is necessary that SLR stabilizes MOC1. The effect on the stability or half-life of MOC1 by SLR1 should be determined under the conditions of inhibition of transcription (such as using actinomycin D) and translation (such as using cycloheximide) experiments.

Response: MOC1 protein levels are increased in both *slr1-d1* and *d18*, and the increase fold is higher in *d18*. This is consistent with the observation that *d18* exhibits longer axillary buds than *slr1-d1* (Supplementary Fig. 3). It is possible that other factors in the downstream of D18 affect MOC1. Moreover, our results showed the MOC1 protein degraded rapidly within 30 minutes and addition of purified His-Trx-SLR1 protein in the cell-free degradation system can dramatically inhibit the degradation of MOC1 but not the His-Trx-GST protein (Fig. 4c). As the secondary transcriptional effect usually happened more than 30 minutes, it is rational to assume that the stabilization of MOC1 was directly caused by SLR1. We have updated the result and added some discussion in the revised manuscript (see highlights, Line 172-179 at page 7 and page 10).

Comment: OK.

Response: Thanks!